# Interference length reveals regularity of crossover placement across species

Marcel Ernst [1,2] ✉, Raphael Mercier [3] & David Zwicker [1] ✉

Crossover interference is a phenomenon that affects the number and positioning of crossovers in meiosis and thus affects genetic diversity and chromosome segregation. Yet, the underlying mechanism is not fully understood, partly because quantification is difficult. To overcome this challenge, we introduce the interference length $L_{int}$ that quantifies changes in crossover patterning due to interference. We show that it faithfully captures known aspects of crossover interference and provides superior statistical power over previous measures such as the interference distance and the gamma shape parameter. We apply our analysis to empirical data and unveil a similar behavior of $L_{int}$ across species, which hints at a common mechanism. A recently proposed coarsening model generally captures these aspects, providing a unified view of crossover interference. Consequently, $L_{int}$ facilitates model refinements and general comparisons between alternative models of crossover interference.

Meiotic crossovers (COs) are crucial for ensuring genetic diversity and are necessary for linking maternal and paternal homologs for proper segregation in most eukaryotes. Chromosomes tend to have at least one CO, but rarely more than a handful. Moreover, CO positions are not independent, but exhibit a phenomenon known as *crossover interference*[1–3]: If chromosomes possess multiple COs, they tend to be spaced more widely than expected by chance. The mechanism governing this CO interference is debated[4–16], in part because it is challenging to quantify CO interference reliably and to compare it across species, mutants, and chromosomes.

COs can be detected in cytology using fluorescent imaging of proteins marking CO sites[7,9,14,17–22]; their position then needs to be determined relative to the synaptonemal complex (SC) on which they reside, leading to CO positions quantified in $\mu$m in SC space. Alternatively, genetic techniques can detect transmission events from parental DNA to offspring to identify COs[7,12,14,23–27]; positions along the chromatids are quantified in units of megabases (Mb) in DNA space. However, only half of the designated COs will become a CO on a selected gamete[7,14]. CO maturation inefficiencies can further contribute to discrepancies between the cytologically and genetically obtained data. Both aspects manifest as a random sub-sampling in

genetic data[28–30]. Moreover, cytological methods typically detect only class I COs, but not the less prevalent class II COs[7,11,13], resulting in a systematic difference.

To quantify CO interference, observed CO counts and positions are summarized using various quantities[13,23,28,31–34]. In the simplest case, one plots the histogram of observed adjacent distances of COs and compares it to the expected distribution without interference; see Fig. 1A. To obtain a single quantity associated with CO interference, the distribution of distances between adjacent COs can be fitted by a Gamma-distribution; The resulting shape parameter $v$ quantifies the evenness of CO distances and is associated with CO interference[7,23,33,35–40]; see Fig. 1A. However, $v$ is sensitive to random sub-sampling[6,28,33,40], it only uses data from chromosomes with at least two COs[41], and it is also influenced by other aspects than interference, in particular the typically heterogeneous distribution of CO positions[33]. An alternative quantification is the coefficient of coincidence (CoC), which measures the ratio of observed frequency of CO pairs to the expected frequency in absence of interference as a function of the CO distance[2,3,14,40,42]; see Fig. 1B. The CoC value is close to 1 when interference is absent, but decreases strongly at short distances when interference is present, reflecting the absence of close double-

[1]Max Planck Institute for Dynamics and Self-Organization, Am Faßberg 17, 37077 Göttingen, Germany. [2]University of Göttingen, Institute for the Dynamics of Complex Systems, Friedrich-Hund-Platz 1, 37077 Göttingen, Germany. [3]Department of Chromosome Biology, Max Planck Institute for Plant Breeding Research, Carl-von-Linné-Weg 10, 50829 Cologne, Germany. ✉e-mail: marcel.ernst@ds.mpg.de; david.zwicker@ds.mpg.de

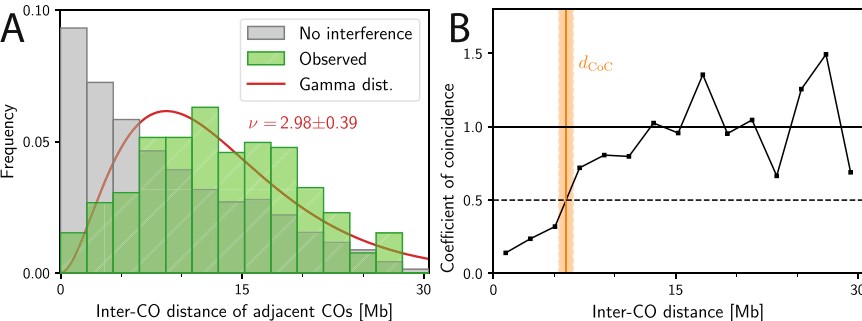

**Fig. 1 | Visualization of traditional quantifications of CO interference.** Shown is genetic data from chromosome 1 of wild-type male *A. thaliana*[12,48]. **A** Comparison of the observed distributions of distances of *adjacent* CO pairs to the expected distribution in the absence of interference (obtained by shuffling all CO positions assuming the same distribution of CO count). The indicated shape parameter $\nu$ follows from a fitted Gamma-distribution (solid line)[7,23,31,33,35–40]; see section 1A of the Supplementary Information. **B** Coefficient of coincidence as a function of the normalized distance between COs[2,3,14,40,42]; see section 1B of the Supplementary Information. The interference distance $d_{\text{CoC}}$ (orange line, shaded area indicates the standard error of the mean) marks the point where the curve first exceeds 0.5.

COs. The distance $d_{\text{CoC}}$ at which the CoC curve crosses 0.5 (orange band in Fig. 1B) provides a length, which tends to be larger for stronger interference[6,13,43,44]. However, this transition point often cannot be located accurately, presumably because it is sensitive to only the data in its vicinity, thus ignoring a potentially large part of the data that could provide information about CO interference. Moreover, the CoC curve relies on binning, which results in information loss[31,33] and requires the difficult choice of an optimal bin count[12,28].

We here introduce the *interference length* $L_{\text{int}}$ to complement previous quantifications. After defining $L_{\text{int}}$ and describing basic properties, we validate it using known behavior of CO interference. We show that $L_{\text{int}}$ can be used to faithfully compare cytological and genetic data from various species, mutants, and chromosomes. Surprisingly, most of these data can be described by a simple normalized interference length, capturing the regularity of CO positions. This suggests a common mechanism underlying CO interference. Indeed, the recently proposed coarsening model[8,9,12,13] explains this behavior qualitatively.

## Results

### Defining interference length as a measure for crossover interference

Crossover (CO) interference is quantified based on the observed CO count per chromosome, $N$, and the associated CO positions $x_i$ along each chromosome. One central quantity is the mean number of COs per bivalent, $\langle N \rangle$, which is typically reduced when CO interference is strong. However, $\langle N \rangle$ does not contain any information about CO positions, so it cannot capture the fact that it is unlikely to find COs in close proximity. To capture such positional information, the main idea of the interference length $L_{\text{int}}$ is to measures the increase of distances between all (not just adjacent) CO pairs due to CO interference. This increase can be expressed by the difference

$$L_{\text{int}} = d_{\text{int}} - d_{\text{noInt}}, \tag{1}$$

where $d_{\text{int}}$ quantifies observed distances, with a correction for variations in the distribution of the CO count $N$, which we introduce in detail below. In contrast, $d_{\text{noInt}}$ quantifies the distance in the null hypothesis without interference. Motivated by the *zyp1*-mutant in *A. thaliana*[12,45], we choose a null hypothesis where COs are placed independently along the chromosomes, sampling from all observed CO positions. In this null hypothesis, the CO count $N$ per chromosome follows a Poisson distribution with the same mean $\langle N \rangle$ as the observed data[42]. We define the associated distance $d_{\text{noInt}}$ as the average distance between any two COs chosen from the pool of all samples for a given chromosome. This definition of $d_{\text{noInt}}$ preserves the CO density along the chromosome.

To quantify the observed distances and define $d_{\text{int}}$, we could have simply used the average distance $d_{\text{obs}}$ of all observed CO pairs. However, this naive choice would only take into account chromosomes with at least two COs, and completely ignore those with one or zero COs. These samples without any CO pairs can represent a large portion of the observation, e.g., in *A. arenosa*[46] and *C. elegans*[5,7,47] or in genetic data from *A. thaliana*[12,48]. In such cases, the naive choice would then only consider data from the small subset with two or more COs, which would dominate the quantity. More importantly, if most samples only carried the obligate CO, strong interference would be likely, which our quantity should capture. These arguments show that the distribution of the observed CO count $N$ per chromosome needs to be considered for defining $d_{\text{int}}$. The observed distribution of CO counts $N$ in case of interference is generally narrower than the Poisson distribution of the null hypothesis of no interference; see Fig. 2A. This deviation, even if it is small, can have a significant impact on the number of observed pairs, because there are $\frac{1}{2}N(N-1)$ pairs for a chromosome with $N$ COs. To see this, imagine observed data of three chromosomes with two COs each, resulting in three distinct pairs; see Fig. 2B. In contrast, without interference, we might have one, two, and three COs on these chromosomes since the distribution of $N$ is broader. This would lead to a total of four possible CO pairs, thus providing more pairs than in the observed data, despite identical $\langle N \rangle$. This example illustrates that the narrower observed distribution of CO counts $N$ leads to fewer CO pairs than the null hypothesis without interference. To account for these *missing pairs*, we compare the average number of observed pairs, $\bar{N}^{\text{pair}}_{\text{obs}}$, to the average number of pairs in the null hypothesis, $\bar{N}^{\text{pair}}_{\text{noInt}} = \frac{1}{2}\langle N \rangle^2$, which follows from the assumed Poisson distribution; see section 2A of the Supplementary Information. The difference quantifies the average number of missing pairs, $\bar{N}^{\text{pair}}_{\text{mis}} = \bar{N}^{\text{pair}}_{\text{noInt}} - \bar{N}^{\text{pair}}_{\text{obs}}$. A larger value of $\bar{N}^{\text{pair}}_{\text{mis}}$ indicates stronger interference, which should be reflected in our measure via a suitable definition of $d_{\text{int}}$.

The distance $d_{\text{int}}$ quantifies the distance of CO pairs in case of interference, which should capture the actually observed distances as well as the fact that interference is stronger when there are more missing CO pairs. We thus define $d_{\text{int}}$ using a weighted average of observed and missing pairs,

$$d_{\text{int}} = \frac{\bar{N}^{\text{pair}}_{\text{obs}} d_{\text{obs}} + \bar{N}^{\text{pair}}_{\text{mis}} d_{\text{mis}}}{\bar{N}^{\text{pair}}_{\text{obs}} + \bar{N}^{\text{pair}}_{\text{mis}}}, \tag{2}$$

where $d_{\text{obs}}$ is the mean distance between all (not just adjacent) CO pairs on the same chromosome. In contrast, $d_{\text{mis}}$ quantifies the distance associated with missing pairs. For simplicity, we assume that $d_{\text{mis}}$ is a constant, and in particular does not depend on the distribution of CO positions. The value of $d_{\text{mis}}$ cannot be larger than the chromosome

length $L$ since such distances can principally not be observed. We thus choose the largest possible value, $d_{mis} = L$, as the most natural length scale; For cytological data, we for simplicity use the average SC length of the respective chromosome, neglecting variations (c.f. ref. 49). We will discuss below how this choice is related to the maximal interference length that can realistically be observed. Taken together, the interference length can be expressed as

$$L_{int} = \phi(d_{obs} - d_{noInt}) + (1 - \phi)(L - d_{noInt}), \qquad (3)$$

where $\phi = \bar{N}^{pair}_{obs}/\bar{N}^{pair}_{noInt} = 2\bar{N}^{pair}_{obs}/\langle N \rangle^2$ denotes the ratio of observed to expected CO pairs, which is small in case of strong interference; compare Fig. 3A. Eq. (3) highlights that the interference length $L_{int}$ combines information of (i) the distribution of CO positions via $d_{noInt}$, (ii) the distribution of the observed distances of CO pairs via $d_{obs}$, and (iii) the distribution of observed CO counts via $\phi$.

Figure 3A shows a graphical interpretation of the interference length $L_{int}$ based on the histogram of the distances between all CO pairs per sample. In contrast to Fig. 1A, we account for missing CO pairs, which contribute with the maximal distance $L$ (cyan region). Consequently, the mean distance of the observed data shifts to larger values (compare dashed green lines in Fig. 1A and Fig. 3A), capturing that missing CO pairs indicate strong interference. Figure 3B visualizes the same idea using cumulative distribution functions. Here, $L_{int}$ corresponds to the blue area between the gray curve representing the null hypothesis and the green curve for observed data with interference, which is scaled by $\phi$ to account for missing CO pairs. The cumulative distribution function highlights that $L_{int}$ can be determined without binning, abolishing this step that could degrade data quality.

The interference length $L_{int}$ has multiple properties that make it a suitable measure of CO interference: (i) $L_{int}$ is a scalar quantity of dimension *length*. Consequently, $L_{int}$ is reported in units of $\mu m$ for cytological data (SC space), and units of megabases (Mb) for genetic data (DNA space). (ii) We show in section 2B of the Supplementary Information that $L_{int}$ is invariant to random sub-sampling (similar to CoC curves), which facilitates the comparison of cytological and genetic data. (iii) $L_{int}$ uses all empirical data on CO positions and does not use any binning or parametrization. On the one hand, all observed CO pairs contribute equally to $d_{obs}$ and thus $L_{int}$; see Eqs. (2)–(3). On the other hand, the definition also accounts for chromosomes without COs or only one CO via the average number of missing pairs, $\bar{N}^{pair}_{mis}$. (iv) The quantity $d_{noInt}$ is based on the observed distribution of CO positions along the chromosome, so that variations of CO density, e.g., due to suppression in centromeric regions, are incorporated in $L_{int}$. (v) $L_{int}$ allows for uncertainty estimations (section 2C of the Supplementary Information) and significance testing (section 2D of the Supplementary Information). We provide a reference implementation of $L_{int}$ with the Supporting Material.

## Large interference lengths indicate strong interference

To see how well the interference length $L_{int}$ captures CO interference, we start by comparing it to the more traditional CoC curves. Figure 3C shows four representative CoC curves for various strains of *A. thaliana*, known to exhibit very different CO interference. In all cases, $L_{int}$ (blue bands) qualitatively captures the distance at which the CoC curve approaches 1, indicating the point at which distances between COs are as frequent as in the null hypothesis without interference. In particular, $L_{int}$ is larger for cases known to exhibit strong interference (e.g., the HEI10[het] mutant), and it correlates (cf. Supplementary Fig. 3C) with the interference distance $d_{CoC}$ (orange band; where the CoC curves exceeds 0.5). However, $L_{int}$ can be calculated more precisely (indicated by the smaller standard error of the mean; see Supplementary Fig. 1B), and it can also be determined for cases without interference (e.g., the *zyp1* mutant) and when few CO pairs are observed. This first analysis

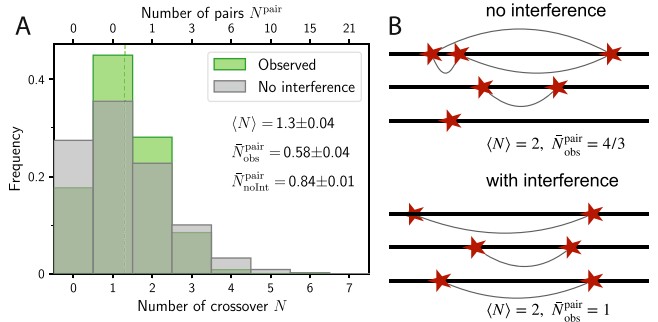

**Fig. 2 | Crossover interference reduces the number of CO pairs. A** Comparison of the observed distribution (green) of the number $N$ of COs per chromosome to the reference without interference (gray) for the same genetic data as in Fig. 1. The corresponding number of CO pairs, $N^{pair} = \frac{1}{2}N(N-1)$, are indicated with respective means. **B** Schematic CO placements on three chromosomes highlighting the effect of interference. The upper panel shows chromosomes with one, two, and three COs, consistent with the broad Poisson distribution in the case without interference. In contrast, interference typically leads to a narrower distribution (bottom panel), where each chromosome has two COs. While both cases have the same mean CO count, $\langle N \rangle = 2$, the thin gray lines indicate that we have a total of three CO pairs with interference ($\bar{N}^{pair}_{obs} = 1$), and thus less than in absence of interference ($\bar{N}^{pair}_{obs} = \frac{4}{3}$), suggesting interference reduces $\bar{N}^{pair}_{obs}$.

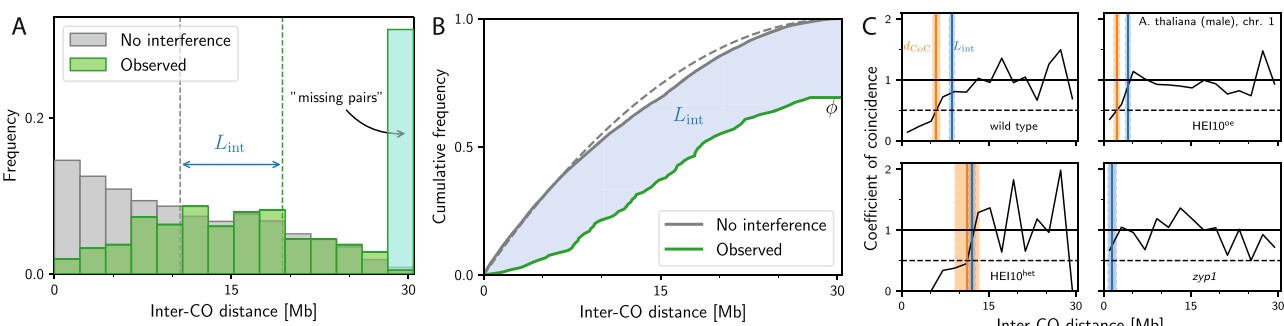

**Fig. 3 | Visualizations of interference length $L_{int}$.** Shown is data for the first chromosome of male meiosis of *A. thaliana*[12,48]. **A** Comparison of the observed (green) and expected (gray) distribution of distances of all CO pairs for wild-type data. The last, cyan bin accounts for missing pairs, which contribute with a length of $d_{mis} = L$ where $L = 30.4$ Mb is the measured chromosome length. The interference length $L_{int}$ is the distance between the mean values of these distributions (denoted by vertical dashed lines). **B** Cumulative distributions visualizing the same data as in panel A. Since the cumulative distribution is an integrated measure, binning is not required and $L_{int}$ corresponds to the blue area. The dashed gray line indicates the theoretical distribution for uniform CO distributions. **C** Coefficient of coincidence curves of four different genotypes of chromosome 1 of male *A. thaliana*[12,48]. Vertical bands mark associated interference lengths $L_{int}$ (blue) and interference distances $d_{CoC}$ (orange) with respective standard error of the mean.

thus indicates that $L_{int}$ captures essential aspects of CoC curves and CO interference.

The only crucial parameter in the definition of $L_{int}$ is the distance $d_{mis}$ associated with missing pairs. Our choice of $d_{mis} = L$ implies that $L_{int}$ assumes values on the order of the chromosome length $L$ in cases of strong interference. Since there are multiple cases that could be called "strong interference", we next evaluate $L_{int}$ for four theoretical scenarios: (i) When all chromosomes exhibit exactly one CO per chromosome, we have $\phi = 0$ and thus $L_{int} = L - d_{noInt}$. In this scenario of *complete interference*, we obtain $L_{int} = \frac{2}{3}L$ when COs are distributed uniformly along the chromosome; see section 2E of the Supplementary Information. These results persist if some chromosomes have no CO instead of one. (ii) We also find $L_{int} = \frac{2}{3}L$ when all chromosomes have exactly two COs at opposite ends of the chromosome. (iii) The *maximal-interference model* of $L_{int}$ for a given average CO count $N$ yields $L_{int} = \frac{4}{3}LN^{-1}$ (limited to $L_{int} = L$ for $N = 1$ when COs always occur at the same position); see section 2F of the Supplementary Information. (iv) Finally, we consider the case where exactly $N$ COs are placed at fixed distance $L/N$, and the first CO is located uniformly between 0 and $L/N$, so the overall CO frequency is uniform along the chromosome. This *regular-placement model* predicts $L_{int} = L[N^{-1} - \frac{1}{3}N^{-2}]$; see section 2G of the Supplementary Information. Taken together, these theoretical scenarios suggest two limiting behaviors of $L_{int}$ in case of strong interference: For few COs, $\langle N \rangle \approx 1$, the first two scenarios suggest $L_{int} \approx \frac{2}{3}L$. Conversely, for many COs, $\langle N \rangle \gg 1$, the last two scenarios suggest the scaling $L_{int} \sim L/\langle N \rangle$. We expect that intermediate values of $\langle N \rangle$ interpolate between these two extremes. In the contrasting case without interference, when the CO count $N$ follows a Poisson distribution and COs are placed independently (but not necessarily uniformly) along the chromosome, we have $\phi = 1$, $d_{obs} = d_{noInt}$, and hence $L_{int} = 0$, corresponding to the null hypothesis without interference. This indicates that larger values of $L_{int}$ are associated with stronger interference, and that the precise value depends on $L$ and $\langle N \rangle$. We next test these predictions for experimental data.

## Interference length recovers sex differences and mutant behavior

We start by using the interference length $L_{int}$ to query known properties of CO interference across different chromosomes, genotypes, and species. Since $L_{int}$ is invariant to sub-sampling, cytological and genetic data can be compared directly, assuming that non-interfering class II COs are negligible and that chromosomes are compacted uniformly. To test this, we took advantage of published data where both genetic and cytological data where available. This includes human male[50,51], as well as *A. thaliana* wild type and mutants with variations in

the expression levels of HEI10[9,12,45,48,52]; see details of data handling in section 3 of the Supplementary Information. Figure 4A shows that the average CO count $\langle N \rangle$ of the cytological data is approximately twice that of the genetic data. This is consistent with expected sub-sampling since a CO detected in cytology affects only two of the four chromatids, and is thus detected in only half the gametes[28–30]. The data also suggests that non-interfering class II COs are negligible, consistent with the low fraction of class II COs in *A. thaliana*, which is estimated at maximally 15%[53,54].

We next compare the interference lengths $L_{int}/L$ determined for the genetic and cytological data normalized with the chromosome length and the SC length, respectively. Figure 4B shows that cytological and genetic data lead to very similar values of $L_{int}/L$. In particular, the null hypothesis that the values agree is not rejected for *A. thaliana* wild type ($p = 0.95$, significance test described in section 2D of the Supplementary Information), HEI10[oe] ($p = 0.49$), HEI10[het] ($p = 0.85$), and *zyp1* ($p = 0.68$), as well as human ($p = 0.10$).

Another important feature of CO interference are sex differences, where CO rates differ between female and male. In *A. thaliana*, female meiosis generally features fewer COs and stronger CO interference according to coefficient of coincidence (CoC) analysis[9,12,21,27]. Figure 4C shows that genetic data[12,48,52] of females indeed exhibit larger interference lengths $L_{int}$ in DNA space than males. This difference is significant in wild type ($p = 10^{-4}$), but not in HEI10[oe] ($p = 0.06$) and in HEI10[het] ($p = 0.32$). It is generally accepted that interference propagates in the $\mu m$ space of the SC[6,21]. Indeed, when we convert the genetic data from DNA space to SC space using the chromosome and SC lengths reported in ref. 12 and then calculate $L_{int}$, the difference between female and male is less significant for *A. thaliana* wild type ($p = 0.02$) and is absent for HEI10[het] ($p = 0.09$) as well as HEI10[oe] ($p = 0.83$) see Fig. 4D. Taken together, this supports a common process in male and female governing CO interference in SC space, whereas sex differences are a consequence of different chromosome organisation, consistent with literature[6,21].

To corroborate this, we also investigated sex differences for human data from cytological imaging of MLH1 foci[50], where CoC analysis in SC space suggest no significant sex difference, thus implying weaker interference for females if measured in DNA space due to lower DNA compaction in female meiosis[50]. Instead, we find a weakly significant difference for $L_{int}$ for the cytological data (Fig. 4D, $p = 0.03$), whereas converting cytological data from SC space to DNA space using chromosome lengths reported in ref. 55 results in significantly smaller $L_{int}$ for females (Fig. 4C, $p = 10^{-6}$). Our analysis again suggests that sex differences are predominantly caused by different chromosome compaction, whereas female and male exhibit similar CO interference in SC space.

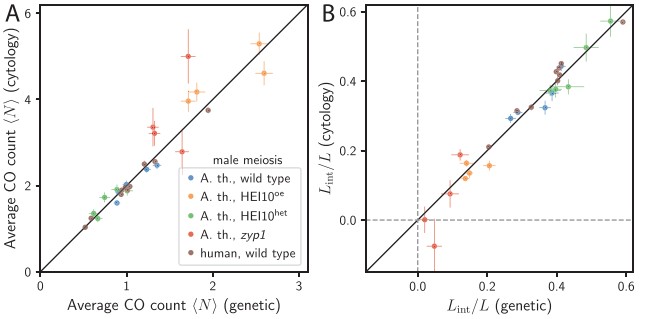
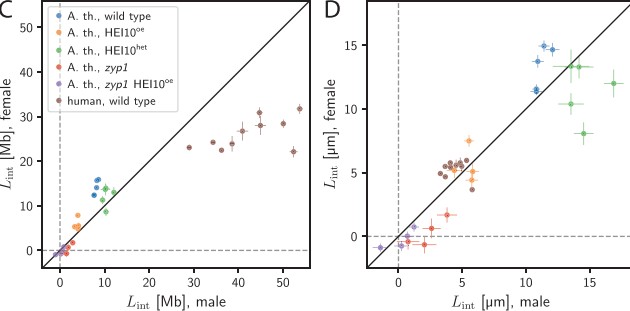
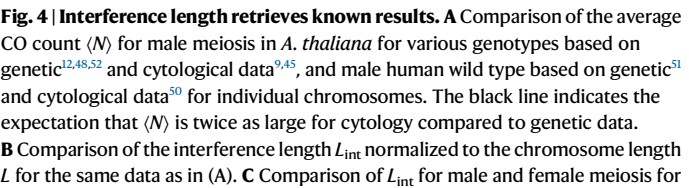

**Fig. 4 | Interference length retrieves known results. A** Comparison of the average CO count $\langle N \rangle$ for male meiosis in *A. thaliana* for various genotypes based on genetic[12,48,52] and cytological data[9,45], and male human wild type based on genetic[51] and cytological data[50] for individual chromosomes. The black line indicates the expectation that $\langle N \rangle$ is twice as large for cytology compared to genetic data. **B** Comparison of the interference length $L_{int}$ normalized to the chromosome length $L$ for the same data as in (A). **C** Comparison of $L_{int}$ for male and female meiosis for

various genotypes based on genetic data of *A. thaliana*[12,48,52] and wild-type, as well as cytological data for human[50] scaled with the respective DNA lengths according to[55] and thus measured in DNA space [Mb]. **D** Comparison of $L_{int}$ of the same data as in **C**; *A. thaliana* data is scaled with respective SC lengths[12,50] and thus measured in SC space [μm]. **A–D** Error bars indicate standard error of the mean. Data handling is detailed in section 3 of the Supplementary Information. Source data are provided as a Source Data file.

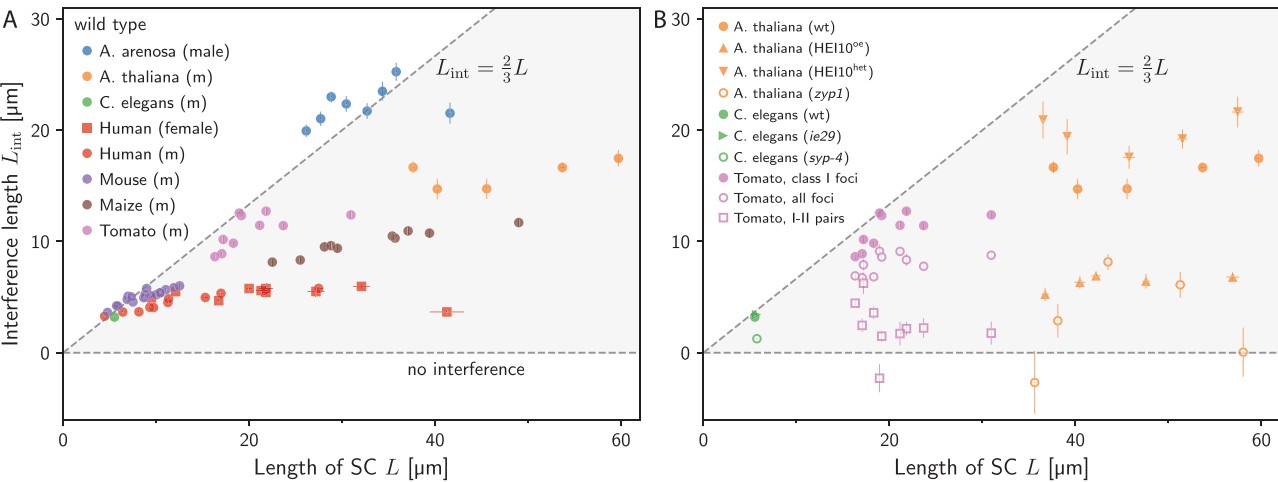

**Fig. 5 | Interference length allows for simple comparison across species and genotypes. A** Interference length $L_{int}$ as a function of SC length $L$ for cytological data of wild-type data of *A. arenosa*[46], *A. thaliana*[9], *C. elegans*[47], human[50], maize[58], mouse[57], and tomato[18]. **B** Interference length $L_{int}$ as a function of SC length $L$ for cytological data of indicated genotypes for *A. thaliana*[9,45], *C. elegans*[47], and tomato[18]. For tomato, we present the interference length of class I COs, of all observed foci (class I and class II CO), as well as pairs with one class I and one class II CO (cf. section 2I of the Supplementary Information). **A–B** Error bars indicate standard error of the mean. Data handling is detailed in section 3 of the Supplementary Information. Source data are provided as a Source Data file. An analogous representation of the genetic data of *A. thaliana*[12,48,52], human[51], and *S. cerevisiae*[59] is given in the Supplementary Fig. 4.

Finally, we test whether $L_{int}$ recovers the behavior of *A. thaliana* mutants. Increasing HEI10 levels (HEI10$^{oe}$ line) decreases $L_{int}$ for both male ($p = 10^{-3}$) as well as female ($p = 4 \cdot 10^{-4}$) in genetic data[12] and for male cytological data ($p = 2 \cdot 10^{-4}$)[46]; see Fig. 4B–D. Lowering HEI10 levels (HEI10$^{het}$ line) increases $L_{int}$ for male genetic data ($p = 0.04$)[12] and cytological data ($p = 0.045$)[46], but $L_{int}$ remains unchanged for female genetic data ($p = 0.15$), suggesting that CO interference is already almost maximal ($L_{int}/L = 0.52...0.67$). For mutants where the SC is absent[12,27,45,56], $L_{int}$ is consistent with absent interference in female *zyp1* mutant[12] ($p = 0.60$), the male *zyp1* mutant (cytology)[45] ($p = 0.23$) and the double mutant *zyp1* HEI10$^{oe}$[12] (male $p = 0.56$, female $p = 0.78$), whereas male *zyp1* mutants (genetic)[12] might exhibit some residual interference (absent with $p = 0.04$). We thus showed that the interference length $L_{int}$ recovers known behavior of CO interference in *A. thaliana* mutants.

**Interference length facilitates comparison across multiple species**

We established that the interference length $L_{int}$ tends to be larger when CO interference is stronger and that this correlation recovers many aspects of CO interference. However, we so far have not interpreted the numeral value of $L_{int}$ in detail, particularly when comparing different genotypes or even different species. Since $L_{int}$ is a single number, such a comparison is easily feasible and can shed light onto the mechanism of CO interference in different species.

To compare measured interference lengths $L_{int}$ of different species, we show $L_{int}$ obtained from cytological data, and thus only interfering class I COs, as a function of the SC length $L$ in Fig. 5A. Evidently, $L_{int}$ can vary widely across species, even when SC lengths are comparable. For instance, for $L \approx 40\,\mu m$, *A. arenosa* exhibits $L_{int} \approx 20\,\mu m$, whereas *A. thaliana*, maize, and human exhibit progressively smaller values down to $L_{int} \approx 5\,\mu m$, suggesting reduced CO interference. However, we also find that $L_{int}$ is correlated with $L$: Multiple species (*A. arenosa*[46], *C. elegans*[47], mouse[57], and tomato[18]) exhibit data very close to the line $L_{int} \approx \frac{2}{3}L$, which we associate with complete interference motivated by the theoretical scenarios studied above. Whereas these species exhibit an almost proportional relationship between $L_{int}$ and $L$, other species (maize[58], *A. thaliana*[9], and human[50]) exhibit a weaker dependence. The associated values of $L_{int}$ are smaller than $\frac{2}{3}L$, indicating incomplete interference. However, all observed

wild-type values are significantly larger than zero, suggesting that they all exhibit CO interference. Taken together, this initial comparison suggests that species either exhibit strong interference close to maximal values ($L_{int} \approx \frac{2}{3}L$) or they exhibit smaller values and weaker $L$-dependence.

We next investigate how mutations change $L_{int}$ for a few species. Figure 5B. shows that the *C. elegans ie29* strain (green triangle) has the same value of $L_{int}$ as the wild type ($p = 0.68$), suggesting that this strain does not exhibit altered CO interference. In contrast, $L_{int}$ is strongly reduced for *C. elegans syp-4* mutants (green circle[47]), consistent with the idea that an intact SC is required for CO interference. We observe a similarly strong reduction of $L_{int}$ in *A. thaliana zyp1* mutants (orange circles), consistent with the described abolished interference[9,45]. In *A. thaliana*, $L_{int}$ can also be reduced by over-expressing HEI10 (orange triangles pointing up), whereas interference is increased when HEI10 levels are reduced in the HEI10$^{het}$ strain (orange triangles pointing down), consistent with the analysis of the genetic data shown above and literature[9,12,45].

A challenge in interpreting CO interference experimentally is that some methods (e.g., based on labeling MLH1 in cytology) only observe class I COs, whereas others (e.g., based on electron microscopy or genetics) cannot distinguish class I COs from class II COs[7,11,13]. A study in tomato[18] used correlative microscopy to identify MLH1-positive recombination nodules (class I CO) and MLH1-negative nodules (class II CO) in the same cells. We analyzed these data and determined $L_{int}$ for various combinations of the two classes of COs; see Fig. 5B. The resulting $L_{int}$ is largest when it is determined only for class I COs (pink disks), which are known to exhibit interference. The value reduces significantly ($p \approx 10^{-4}$) when $L_{int}$ is calculated based on all foci (pink circles), and this reduction is consistent ($p = 0.29$) with an approximate correction of $L_{int}$ taking class II COs (6% to 19%) into account; see section 2H of the Supplementary Information. We also quantify how class II COs interfere with the positioning of class I COs by evaluating $L_{int}$ associated with pairs comprising a class I CO and a class II CO (pink squares); see section 2I of the Supplementary Information. These mixed pairs exhibit a positive ($p = 0.01$), weaker interference ($p = 10^{-4}$, compared with $L_{int}$ of all foci), but we have no evidence that class II COs interfere with each other ($p = 0.58$, $L_{int}$ not shown in figure) or exhibit different interference than the mixed pairs ($p = 0.51$), consistent with literature[18].

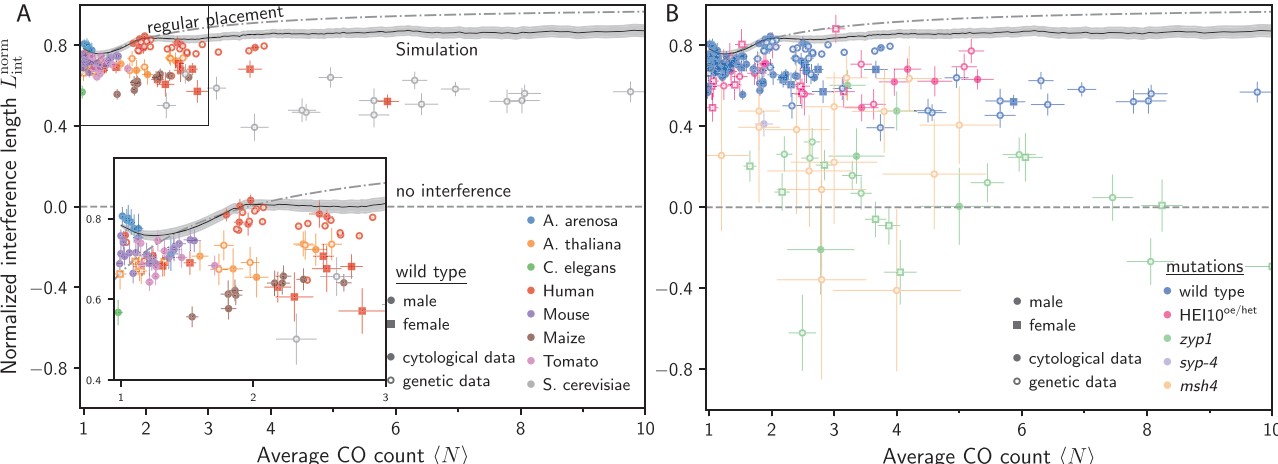

**Fig. 6 | Normalized interference length unveils similarity of mutant behavior across species. A** Normalized interference length $L_{int}^{norm} = L_{int}\langle N \rangle / L$ as a function of the mean CO count $\langle N \rangle$ for wild-type data of *A. arenosa*[46], *A. thaliana*[9,12,48], *C. elegans*[47], human[50,51], maize[58], mouse[57], tomato[18], and *S. cerevisiae*[59] using both cytological and genetic data. **B** $L_{int}^{norm}$ as a function of $\langle N \rangle$ for the same wild-type data as in Panel A (violet), mutations with altered HEI10 levels in *A. thaliana* (magenta,[9,12,52]), mutations that affect the SC in *A. thaliana* and *C. elegans*

(orange,[12,45,47]) and the *msh4* mutant for *S. cerevisiae* (gold,[59,60]). **A**–**B** The dashed line marks the prediction of the regular-placement model corresponding to strong interference (see section 2G of the Supplementary Information), whereas the black line corresponds to the coarsening model for *A. thaliana*[12]. Error bars indicate standard error of the mean. Data handling is detailed in section 3 of the Supplementary Information. Source data are provided as a Source Data file.

Taken together, these data show that the interference length $L_{int}$ recovers central observations about CO interference. In particular, values of $L_{int}$ tend to vary between small values (indicating absence of interference) and large values $L_{int} \approx \frac{2}{3}L$ (indicating strong interference). While we briefly explored the dependence on the chromosome length $L$, we expect from our theoretical analysis that $L_{int}$ also depends on the mean CO count $\langle N \rangle$, which could distort the interpretations we made so far.

**Crossover interference exhibits similarity across species and mutants with intact SC**

The maximal-interference model and the regular-placement model suggest the scaling $L_{int} \sim L/\langle N \rangle$, i.e., that $L_{int}$ is generally larger for longer chromosomes and fewer COs. To test this scaling, we analyze the normalized interference length, $L_{int}^{norm} = L_{int}\langle N \rangle / L$, which would be a constant if the scaling held perfectly. Since $L/\langle N \rangle$ estimates the expected distance between COs, $L_{int}^{norm}$ relates to the regularity of COs placement along chromosomes. Note that $\langle N \rangle$ is the number of COs per bivalent, implying that we need to double the CO counts measured for individual chromatids in genetic data to account for the subsampling. Figure 6A shows that $L_{int}^{norm}$ clusters around values between ~0.6 and ~0.8 for wild types of many species, particularly when they have few COs ($\langle N \rangle \lesssim 4$). Notable exceptions are *C. elegans*, which exhibits a skewed distributions of CO positions, and *S. cerevisiae*, which generally seems to exhibit weaker CO interference than other species we analyzed[13,43].

To explain the observed narrow band of $L_{int}^{norm}$, we compare the data to two theoretical predictions. First, we investigate the regular-placement model (dashed lines in Fig. 6), where $\langle N \rangle$ COs are placed uniformly with separation $L/\langle N \rangle$. This model overestimates $L_{int}$ for larger values of $\langle N \rangle$, likely because CO placement is not as regular in reality. Interestingly, the model underestimates $L_{int}$ for small $\langle N \rangle$, which is a consequence of its uniform CO placement along the chromosome, whereas the observed distributions are often highly non-uniform. Second, we study the recently proposed coarsening model of CO interference[8,9,12] for parameters obtained for *A. thaliana*[12]. While the model (black lines in Fig. 6) captures the general trend better than the regular-placement model, there are significant deviations: The model overestimates $L_{int}^{norm}$ for most species, except *A. arenosa*[46], most likely because of very localized CO positions. The discrepancies

between data and model revealed by $L_{int}^{norm}$ could guide future model refinements.

Our analysis of the normalized interference length $L_{int}^{norm}$ for simple models and wild-type data suggests that systems with strong interference exhibit similar values of $L_{int}^{norm}$, which depend only weakly on $L$ and $\langle N \rangle$. In particular, the normalized interference length removes the dependency on $L$ and $\langle N \rangle$ that dominated in Fig. 5A: On the one hand, the species obeying the scaling $L_{int} \sim \frac{2}{3}L$ all exhibit $\langle N \rangle \approx 1$, implying that the associated $L_{int}^{norm}$ is roughly 0.7. On the other hand, the cases in Fig. 5A that deviated from this scaling all exhibit more COs, explaining the reduced values of $L_{int}$. Consequently, all the cases shown in Fig. 5A (except human female chromosome 1 with $\langle N \rangle \approx 6$) exhibit values of $L_{int}^{norm}$ between ~0.6 and ~0.8. This similarity in $L_{int}^{norm}$ in all analyzed species (except *S. cerevisiae*, which exhibits larger CO counts and lower $L_{int}$) indicates a similar regularity in CO placement, which could originate from a similar mechanism that governs CO interference in these different species.

We next test the hypothesis that the normalized interference length $L_{int}^{norm}$ captures an essential aspect of the CO interference process by comparing wild-type data with mutants known to affect CO interference. Figure 6B shows that mutants affecting the SC (orange and gold symbols) exhibit lower values of $L_{int}^{norm}$, distributed around $L_{int}^{norm} = 0$. This observation is consistent with the strongly reduced interference described in the literature[12,45,47,59,60], which disrupts the regularity of CO placement. In contrast, *A. thaliana* mutants with altered HEI10 levels (magenta symbols) exhibit values of $L_{int}^{norm}$ that are consistent with the wild-type results (violet symbols). Apparently, changing HEI10 levels only affects the CO count $\langle N \rangle$ but not the CO interference as measured by $L_{int}^{norm}$. Taken together, we propose that $L_{int}^{norm}$ quantifies aspects of CO interference that are independent of $\langle N \rangle$, which suggests that CO interference is not affected by changing HEI10 levels, but is strongly impaired in mutants affecting the SC. This interpretation is consistent with the coarsening model, where the SC is vital for mediating coarsening between COs on the same chromosome, whereas changing HEI10 levels merely affects the degree of coarsening without disrupting the mechanism.

## Discussion

In this paper we propose the *interference length $L_{int}$*, which summarizes deviations in CO placement, as a quantity to measure CO interference.

$L_{int}$ is a physical length, which is larger for stronger CO interference and reaches a maximum of about $0.8L$ in empirical data. The fact that $L_{int}$ provides a single number to measure CO interference enables direct comparison of data for different chromosomes, genotypes, and species among each other and with theoretical models. The quantity is also invariant to random sub-sampling (enabling comparison of genetic and cytological data), does not require binning, and uses all empirical data, particularly those from chromosomes with one or no COs. Supplementary Table 1 compares these features to the alternative quantities $d_{CoC}$ and $\nu$. A distinct advantage of $L_{int}$ is the lower uncertainty compared to $d_{CoC}$ and $\nu$ (see Supplementary Fig. 1B), providing more statistical power at the same sample size or allowing for fewer experiments to draw conclusions.

We used $L_{int}$ to query known behavior of CO interference using published data. The comparison across species revealed that $L_{int}$ only reaches maximal values when there are few COs ($L_{int} \approx \frac{2}{3}L$). In contrast, species with larger CO counts typically exhibit smaller values of $L_{int}$, which also vary less with the respective lengths of the chromosomes. This behavior is consistent with the recently proposed coarsening model[8,9,12,13], which suggests that $L_{int}$ typically scales inversely with the CO count $\langle N \rangle$, unless there are few COs and $L_{int}$ saturates; see Supplementary Fig. 2. This suggests that species with many COs simply aborted coarsening before completion, implying larger $\langle N \rangle$ and leading to values of $L_{int}$ that are independent of $L$, whereas species with completed coarsening exhibit only the obligate CO and $L_{int} \approx \frac{2}{3}L$. This strong connection between $\langle N \rangle$ and $L_{int}$ also explains the observed narrow band of values of the normalized interference length $L_{int}^{norm} = L_{int}\langle N \rangle / L$ for cases where coarsening can proceed normally (e.g., in wild type and in HEI10 mutants). A similar analysis using $d_{CoC}$ and $\nu$ does not yield a consistent picture (see Supplementary Fig. 3), suggesting that only $L_{int}$ captures an essential property of CO interference that is nearly preserved across species.

The interference length $L_{int}$ enables comparison of various models with experimental data. In particular, it allows to compare the coarsening model with alternatives, like the beam-film model[28,61]. It will also prove useful to quantify the influence of model parameters onto CO interference. At the same time, $L_{int}$ can also support experimental work, particularly by allowing comparisons between chromosomes, genotypes, and species.

### Reporting summary

Further information on research design is available in the Nature Portfolio Reporting Summary linked to this article.

## Data availability

All data supporting the findings of this study are available within the paper and its Supplementary Information. Source data are provided with this paper.

## Code availability

The code of this study is available via the github repository at https://github.com/zwicker-group/crossover-interference-length[62].

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

## Acknowledgements

We gratefully acknowledge funding from the Max Planck Society and the European Union (ERC, EmulSim, 101044662).

## Author contributions

M.E., R.M., and D.Z. conceived the project and developed the quantification. M.E. investigated its properties and analyzed the data. M.E. and D.Z. wrote the first draft of the manuscript, which all authors edited and approved.

## Funding

## Competing interests

The authors declare no competing interests.
