## [Peer Review File · Nature Communications]

Interference Length reveals regularity of crossover placement across speciesREVIEWER COMMENTS

Reviewer #1 (Remarks to the Author):

Summary:

The authors introduce a new statistic for describing crossover interference, the “interference length” Lint. The authors construct Lint in such a way as to include instances where 0-1 crossovers are formed. They describe its derivation and confirm its behavior for representative model distributions. By reanalyzing a dataset previously described by the authors and analyzed using conventional metrics (the coefficient of coincidence, CoC) (Durand 2022), they show that Lint recapitulates the same trends. They show for several wildtype and mutant datasets that Lint quantitatively scales with the CoC. Applying Lint analysis to multiple published datasets (*A. arenosa*, *C. elegans*, *H. sapiens*, *M. musculus*, *Z. mays*, *S. lycopersicum*, and *S. cerevisiae*), they show Lint accurately describes previously known relationships, e.g. showing that Lint scales with physical (microns) distance between sexes with different synaptonemal complex lengths. Normalizing Lint by distance and by average number of crossovers, they find that Lint-norm falls into a relatively small range of values (~0.6-0.8) across species, that alterations in dosage of the pro-crossover factor HEI10 do not dramatically change this value, and that mutations that alter synaptonemal complex proteins do change this value. (They also find that mutation of the pro-crossover factor MSH4 in yeast results in comparable changes as the SC mutants in *C. elegans* and *A. thaliana*.) While ambiguous, this is promising, and suggests that future work based on such metrics may be able to differentiate changes in crossover interference resulting from changes in crossover number from changes in inherent “crossover strength.”

Overall, the Lint statistic is clearly appropriate for analysis of interference data and is an important update to the CoC (and fits to a gamma distribution): while the CoC was defined over a century ago based on its ability to fit genetic data, Lint is explicitly constructed to analyze spatial data, which we now know is the relevant dimension for interference. It is likely to be used widely by researchers working on crossover interference in a range of systems. Because of its greater precision and robustness to low-crossover and low-interference scenarios, it is likely to improve the investigation of quantitative phenomena and more finely distinguish between potential mechanisms of crossover regulation. Because its logic is clearly defined and mathematically well-founded, it will be amenable to further improvement by the authors or by other researchers.

Feedback:

The Lint metric is based on comparison to a null hypothesis of no interference that has a Poisson distribution of crossovers. However, this does not reflect real systems, which generally bear an obligate crossover regardless of interference strength: these are two incompletely overlapping mechanisms. The authors should justify this choice and what it means for interpretation.

To include data from chromosomes with 0 or 1 crossover, the authors introduce analysis based on “missing pairs.” As the authors note, the definition of d_{mis} is critical because its contribution to Lint directly scales with the number of missing pairs (i.e. $1-\phi$). However, the choice of $d_{mis} = L$, while reasonable, is arbitrary. The authors should further describe the rationale behind this choice.

Related, the authors should report values of phi across the analyzed datasets to help evaluate the impact of this contribution on reported values and for the improvement in precision.

I am confused by the data in Fig. 2A that shows ~18% of chromosomes bear no crossovers. This is in conflict with the obligate crossover assumption. Looking at Durand 2022, it seems like these values may be per-chromatid. However, isn't the relevant metric for interference the crossovers formed along each stretch of SC? Please clarify.

I am grateful that the authors include the msh4 data despite it not clearly fitting their framework of SC vs. crossover factor perturbation. However, they should briefly note this and possibly speculate about possible explanations in the text.

While the authors correctly note that their results are consistent with the coarsening model, alternate plausible models based on the chromatin or SC itself can produce results quantitatively consistent with crossover outcomes (e.g. Zhang PLoS Genetics 2014). The existence of alternate models (and ideally, whether/how Lint or other metrics could in the future be used to analyze them or discriminate between alternatives) should be briefly noted in the text.

Suggestions for presentation:

Some colors/fit lines in Fig S1 either do not match the caption or are missing.

It is difficult to distinguish between members of the orange and of the purple families of colors in Fig 6B and S3.

To improve uptake across the community and help the reader get started with the provided code, I would recommend providing a schematic or pseudocode of how Lint is calculated from data as part of the supplementary material. However, this is not critical.

Reviewer #1 (Remarks on code availability):

Only briefly reviewed. In general, the raw code seems reasonable and brief enough to be parsed by the reader. As hosted, `measure_CO_interference.html` did not produce results (clicking "calculate quantities" after loading the example data gave no response on Chrome or Firefox).

Reviewer #2 (Remarks to the Author):

This article derives two new metrics to measure aspects of crossover interference. While they in some ways provide similar information to L.coc (L.int) and the gamma shape parameter (L.norm.int) their main advantage lies in the fact that they include data from all COs, account for chromosomes with 1 or no COs, do not require any binning or parametrization and are not affected by sub-sampling. As a result, the derived metrics show improved precision over L.coc and gamma.

The determination of the null expectation (d.noint) from observed CO positions also means that non-uniform crossover distributions are taken into consideration.

This work will be of interest to the meiosis community and is very timely, as there is considerable interest currently in modelling crossover interference and testing several competing models. This new metric will likely assist in the interrogation of these models.

There are now quite a few metrics of interference, each of which has their pros and cons. This metric too, has its pros and cons. It has numerous advantages (outlined above) in terms of its calculation and making maximal use of the available data – though these do appear to introduce their own issues e.g. the somewhat arbitrary scaling of L.int with L when interference is strong. On balance though, I think it is a very good contribution to the field and will likely gain use over time.

In general, the article is quite well written, though I have suggested several amendments below to improve understanding and readability.

Methodology is largely sound (with some points listed below), though it would be good to provide some more detail on some of the assumptions and strategy for dealing with cytological data (where specific chromosomes are usually not identified) to aid reproducibility (see below).

Specific points:

Abstract: “We show that it faithfully captures known aspects of crossover interference and provides superior statistical power over previous methods.” As far as I can see only one statistical comparison to previous method is provided in the main body of the text. If multiple statistical comparisons are referenced in the abstract, then they should all be mentioned – and ideally covered in some depth – in the main text.

Line 39: Consider re-wording. “Bias” is somewhat pejorative here. Cytological data and genetic data provide different information. Depending on your interest, one or the other set of data (and the interference metrics derived from them) may be more or less useful. For understanding the mechanistic basis of crossover interference, cytological data (which lacks data from non-interfering COs) is likely to be more informative. For understanding patterns of inheritance, then a metric that takes into account non-interfering crossovers is likely to be of more interest.

Line 101+: You state “d.int quantifies observed distances”, I think this needs some further elaboration here, even if full details are given further down in the text. Presumably this is some form of normalised average inter CO distance?

Line 128: How then are chromosomes with 0 or 1 CO taken into account? The next paragraph needs to be better linked so that the reader knows this information is coming.

183 / Figure 3B: I do not fully understand how to interpret this figure. As a non-mathematician it is not intuitive to me why the blue area between the two cumulative distributions corresponds to

L.int. Some more detail is needed – I imagine many of the researchers most interested in this publication will also not be mathematicians...

Line 243: SI-2E should this be SI-2F?

It is unclear to me if L.int can incorporate data from different chromosomes where d.mis differs or must L.int be calculated independently for different chromosomes. How is d.mis dealt with for cytological data where different chromosomes have different lengths but are not uniquely identifiable? Or when even the same chromosome will usually have different lengths in different cells.

In Figure 4B, I am very surprised that similar values are obtained for L.int/L genetic and L.int/L cytology, given that the latter only includes non-interfering (e.g. MLH1 marked) COs, while the former includes both interfering and non-interfering crossovers. I would have expected L.int/L genetic to be systematically lower than L.int/L cytology. Indeed, this seems to be the implication given the tomato data shown in figure 5B. In this case, L.int/L would be systematically lower when including both class I and class II crossovers. Why then are these estimates equivalent? Has some correction been made to account for class II crossovers in this comparison e.g. Supp 2H?

If a correction has been applied, then

- a) this needs to be made much more explicit, as the results as presented are very misleading
- b) the comparison of uncorrected genetic / cytological estimates needs to be shown instead of (or in addition to) the corrected comparison

If a correction has not been applied, then can the authors provide some explanation as to why the two estimates are not different and why this appears to contrast with the tomato data shown in figure 5B. Is this something to do with subsampling for genetic data (but not for the cytological class II data from tomato)? Or something to do with the different backgrounds? E.g. pure inbred lines for cytology vs hybrids for genetic data? I notice in Figure 4A that crossover counts from genetic data are not systematically higher than crossover counts for cytological data...

It strikes me that there are many assumptions going on here that require some further explanation/consideration in the text. For example, the comparison Fig 4B requires cytological data to be matched to specific chromosomes. My understanding is that in the published data the specific chromosomes on which crossovers are formed have not been identified. How then is this choice then made? Presumably the longest chromosome measured by μm SC is always assumed to the longest chromosome as measured in Mb? Similarly, L.int takes CO positioning into account (when determining d.noint) which has polarity when analysing genetic data based on the reference chromosome sequence. But presumably does not for cytological data where this would require specific labelling of one of the chromosome ends. How is this handled? Is the end which represents the start position randomly chosen?

What is d.mis for cytological data? Does it take into account the fact that there is variation in total SC length between cells? E.g. by averaging the lengths of only chromosomes with 0 or 1 CO to give d.mis?

Line 372-378: One question I had when reading this section was “Is this difference associated with proportion of total crossovers contributed by non-interfering class II COs?” I then realised this is exclusively cytological data and so is class I only. The authors may like to flag up this point somewhere in this paragraph.

Fig 5: The fact that $L.int$ scales with L is somewhat of a drawback to the metric as I assume (and other modelling suggests) that mechanistically, interference is implemented over the same distance of SC on all chromosomes within a cell. Is this scaling primarily due to the fact that $d.mis = L$?

Minor point – what are used as the proxies of SC length in mutants (such as *zyp1*) which lack an SC? Is this just axis length?

Figure 6A: I find there is far too much going on in this Figure I would prefer four separate panels with more of a zoom / of the box in the top left corner (for both A and B), so that the data can actually be seen. In the text a point is made about relative fits of simulations for low N . It is very hard to assess this with the given plots.

Fig 6B: the colours are very similar for *zyp1* and *msh4*. Alternative colours should be used so that these can be distinguished.

It is interesting here that *C. elegans* has a low(ish) value of $L.norm.int$ given complete interference in this species. Presumably for $N \sim 1$, then $\phi = 0$ so $L.int = L - d.noint$. Why is there such variation in $L.norm.int$ for cases where $N \sim 1$? Is this entirely due to differences in CO distribution causing different values of $d.noint$?

This seems to me a limitation of the metric. I.e. quite different values of $L.norm.int$ for different cases of strong interference.

I imagine (though am happy to be corrected!) that a draw back of the $L.int$ metric is that the same value could be derived from situations where only a few COs occur but there is no interference, and situations where many more crossovers occur but interference is present. Where would *msh4* data (few COs, no interference) be, were it shown on figure 5?

Line 461: Is this both for cytological and genetic data?

My take here is that $L.norm.int$ is akin to the gamma model where it tells us about the regularity of inter CO distance, irrespective of absolute inter-crossover distance? I would find a comparative table useful which lays out the respective attributes of gamma, $L.coc$, $L.int$, $L.norm.int$ and their interpretation.

Line 500: “cluster” is quite generous here... the values of $L.norm.int$ appear highly variable in these mutant contexts.

Other points:

Discussion of the contribution to / effect of class II crossovers on L.int and L.norm.int seems limited. It would be good to see how L.int and L.norm.int respond in situations where there are high levels of class II COs (e.g. recq4 mutants in Arabidopsis). Similarly, I think the manuscript would benefit from running simulations to explore how L.int/ L.norm.int respond to differing proportions of class II crossovers e.g. coarsening simulations with different added proportions of randomly distributed COs.

A somewhat leftfield thought, but I wondered if L.int is able to give any meaningful output from F2 data? It would clearly underestimate L.int and be sex-averaged, but perhaps it might respond in a predictable manner (e.g. half the sex average - this could be tested with simulations / theory) and might possibly enable some insights into interference to be gained when backcross data are not available. If so, this would give added strength to the utility of this new metric.

There is some inconsistency in which data are shown for which analyses. E.g. for some both cytological and genetic are shown, for others just cytological (e.g. Fig 5). Some mutants are shown in some figures/analyses but not in others. There are some legitimate reasons for this, even if simply to prevent charts from becoming overcrowded. But to avoid any sense of “cherry-picking” which data are shown, it might be better to have (even if just added to Supp Material) all analyses/charts shown for all data used in the manuscript.

As mentioned above, I think it would be useful to have a comparative table that lays out the respective attributes of gamma, L.coc, L.int, L.norm.int and their interpretation, pros/cons e.g. affected by sub-sampling, etc, etc

Reviewer #2 (Remarks on code availability):

I have tried using the code supplied via Code Ocean. I was able to get the html page working and this seemed to give good results. I could not get the python script to work - perhaps a README file would be useful here. A quick scan suggests it is written specifically for Arabidopsis thaliana data. If so, a general use version may be more helpful.

Reviewer #3 (Remarks to the Author):

During meiosis, crossovers promote reciprocal DNA exchange between paired homologous chromosomes. Consequently, the magnitude of DNA exchange is determined by the position and number of chromosomal crossovers, which are directly influenced by the conserved phenomena of crossover interference (the inhibition of closely-spaced crossovers). Despite crossover interference being studied for over a century, the meiosis field has yet to reach a consensus on a standard approach for reliably quantifying crossover interference. Here, Ernst, Mercier and Zwicker point out inadequacies in current methods for quantifying interference (CoC and gamma shape parameter

analysis) and formulate a new metric (Interference Length) that more comprehensively captures a wide variety of meiotic crossover patterning features in a single number.

The authors describe in detail how the Interference Length metric has been derived and clearly justify the rationale for including particular values. They then demonstrate and explore the functionality of Interference Length by comparing values obtained using publicly available cytological and genetic datasets in different organisms and genetic backgrounds. They conclude that Interference Length offers a useful method (superior to CoC or gamma) for quantitatively comparing aspects of crossover interference both within and across species. Finally, they discuss how inferences drawn from these Interference Length comparisons are consistent with the recently proposed coarsening model for meiotic crossover patterning.

I am convinced that Interference Length offers a superior method for quantifying crossover interference in a single unit compared with previous methods and would be a useful addition to the meiosis field. However, I have a few comments that it would be useful for the authors to address:

Major

1. The authors demonstrate that Interference Length can be used to compare crossover patterning data between organisms and how this could be used, for example, to support a common underlying mechanism (e.g. the coarsening model). However, the calculation required to produce an Interference Length value is complex and it is not always intuitive how this value should be directly interpreted with regard to the crossover patterning factors that directly underlie changes in the value (i.e. why are normalised interference length values in *S. cerevisiae* lower than in other species?). Therefore, the case of *S. cerevisiae* would benefit from further discussion and analysis of why this species is an exception (presumably this is a consequence of the chromosomes being very small and having a high crossover frequency?). I believe cytological Zip3 foci data is available for three *cerevisiae* chromosomes in Ref 43. Perhaps including this data in Fig 5 and 6 would also help to clarify the cause of this difference.

2. I'm still not completely convinced by the authors' claims that "cytological and genetic data can be compared directly" using Interference Length. I understand that Interference Length is invariant to subsampling, but I think there are other points to consider to enable direct comparison. Surely the presence of heterochiasmy (and the authors analysis of this) argues against the suggestion that cytological data and genetic data can be directly compared? The authors plot normalised interference length from genetic and cytological data in Fig 4B and demonstrate agreement for human male interference length. However, if they were able to add human female data to the plot in Fig 4B I doubt this would agree as there would be a change in the normalisation factor for SC length but not for chromosome length. Similarly, plot Fig 4C and Fig 4D demonstrate that measuring Interference Length using genetic data or cytological data can result in very different interpretations (Interference Length is smaller when measured genetically for human males compared with females, but equivalent when measured cytologically). Am I also correct in thinking that genetic Interference Length values in Fig 4 have not been corrected for the presence of class II COs (which as the authors' mention, is a hurdle for interpreting interference in genetic CO data)? On a related

note, is the genetic data in Fig 6 scaled with respective SC lengths (and thus measured in SC space) to enable comparison with cytological data? I would have thought the scaling of genetic data (or not) would influence the impact of heterochiasmy in comparing male and female genetic data in this plot. A final point to note is that the scaling of genetic data by SC lengths assumes that compaction of chromatin along meiotic pachytene chromosomes is uniform. However, this is not the case as compaction (i.e. μm of SC per Mb of DNA) varies significantly between e.g. chromosome arms and pericentromeric regions.

3. It is surprising that the normalised Interference Length values for arabidopsis deviate so much from the predicted values from the coarsening model (Fig 6), given that the coarsening model was parameterised using arabidopsis data. I think further exploration of this deviation is warranted (especially as the authors conclude that their data is consistent with the coarsening model). What parameters within the model would the authors alter to achieve a better fit of the simulated Interference Length to the experimental arabidopsis data? Do simulation outputs from other versions of the model (e.g. from Ref 9) result in better fits to the data? Furthermore, the authors use the parameterisation from arabidopsis to compare to data from many other organisms (Fig 6). There is no reason to think that the Arabidopsis parameters will be universal. Hence, for this comparison to be rigorous, the authors really need to make some attempt to refit the coarsening model for these other cases. This might lead to better agreement between the model and the data.

4. In the authors' analysis, they set the distance associated with missing pairs $d_{\text{mis}}=L$, the chromosome length. While this scaling is reasonable, setting the proportionality constant strictly to one is arbitrary. I wonder whether better results could be obtained using $d_{\text{mis}}=\alpha L$, with α a free parameter.

Minor

5. The authors discuss the benefits of Interference Length compared with CoC and gamma analysis, which both provide a single convenient metric for quantifying interference. However, it is also possible to plot and measure individual aspects of CO patterning (CO number, CO spacing, CO position) separately and then compare these (very easy to understand and interpret) plots individually without the need to compress the information into a single number. I think the manuscript would also benefit from a discussion of why Interference Length offers a superior (or useful complementary) approach compared to analysing individual crossover patterning features in isolation.

6. Line 546 -547. The authors discuss how species with many COs might abort coarsening early. However, there are other explanations, such as changes in the dosage of HEI10 (orthologs), or altered binding/unbinding kinetics. I think these should also be discussed/mentioned.

Reviewer #4 (Remarks to the Author):

1 **Rebuttal Letter: Interference Length reveals regularity of crossover placement across** 2 **species**

Marcel Ernst, Raphael Mercier, and David Zwicker
(Dated: July 12, 2024)

We thank the editor and the four referees for handling our manuscript and giving thoughtful comments. Below we
provide a point-by-point response (blue text) to all the referees' comments (black text). We also mention how we
revised the manuscript and provide a PDF highlighting the differences to the previously submitted version.

**CONTENTS**

I. Reviewer #1 (Remarks to the Author)	1
10	A. Summary:	1
B. Feedback:	2
C. Suggestions for presentation:	3
D. Reviewer #1 (Remarks on code availability):	3
II. Reviewer #2 (Remarks to the Author):	3
15	A. Specific points:	4
B. Reviewer #2 (Remarks on code availability):	7
III. Reviewer #3 (Remarks to the Author):	7
18	A. Major	8
B. Minor	10
IV. Reviewer #4 (Remarks to the Author):	10
References	10

**I. REVIEWER #1 (REMARKS TO THE AUTHOR)**

**A. Summary:**

The authors introduce a new statistic for describing crossover interference, the “interference length” Lint. The
authors construct Lint in such a way as to include instances where 0 – 1 crossovers are formed. They describe
its derivation and confirm its behavior for representative model distributions. By reanalyzing a dataset previously
described by the authors and analyzed using conventional metrics (the coefficient of coincidence, CoC) (Durand 2022),
they show that Lint recapitulates the same trends. They show for several wildtype and mutant datasets that Lint
quantitatively scales with the CoC. Applying Lint analysis to multiple published datasets (*A. arenosa*, *C. elegans*, *H.*
*sapiens*, *M. musculus*, *Z. mays*, *S. lycopersicum*, and *S. cerevisiae*), they show Lint accurately describes previously
known relationships, e.g. showing that Lint scales with physical (microns) distance between sexes with different
synaptonemal complex lengths. Normalizing Lint by distance and by average number of crossovers, they find that
Lint-norm falls into a relatively small range of values ($\approx 0.6 - 0.8$) across species, that alterations in dosage of
the pro-crossover factor HEI10 do not dramatically change this value, and that mutations that alter synaptonemal
complex proteins do change this value. (They also find that mutation of the pro-crossover factor MSH4 in yeast
results in comparable changes as the SC mutants in *C. elegans* and *A. thaliana*.) While ambiguous, this is promising,
and suggests that future work based on such metrics may be able to differentiate changes in crossover interference
resulting from changes in crossover number from changes in inherent “crossover strength.”

Overall, the Lint statistic is clearly appropriate for analysis of interference data and is an important update to
the CoC (and fits to a gamma distribution): while the CoC was defined over a century ago based on its ability to
fit genetic data, Lint is explicitly constructed to analyze spatial data, which we now know is the relevant dimension
for interference. It is likely to be used widely by researchers working on crossover interference in a range of systems.
Because of its greater precision and robustness to low-crossover and low-interference scenarios, it is likely to improve
the investigation of quantitative phenomena and more finely distinguish between potential mechanisms of crossover

regulation. Because its logic is clearly defined and mathematically well-founded, it will be amenable to further
 improvement by the authors or by other researchers.

We thank the referee for their summary of our work and the positive assessment.

B. Feedback:

The Lint metric is based on comparison to a null hypothesis of no interference that has a Poisson distribution
 of crossovers. However, this does not reflect real systems, which generally bear an obligate crossover regardless of
 interference strength: these are two incompletely overlapping mechanisms. The authors should justify this choice and
 what it means for interpretation.

We agree that our null hypothesis corresponds to a particular choice, and does not possess obligate crossovers.
 We choose this null hypothesis since it is similar to the *zyp1* mutant in *A. thaliana*, which does not exhibit CO
 interference and also does not show obligate crossovers [1, 2]. Since we do not know any example that exhibits an
 obligate crossover but no CO interference, we cannot construct an associated null hypothesis for such systems. We
 revised the main text to clarify our choice of the null hypothesis.

In terms of interpretation, we agree that our measure is influenced by the presence of the obligate CO as well as
 CO interference, and these effects cannot be clearly separated. However, both the coarsening model [1, 3] and beam-
 film model [4, 5] suggest that both phenomena originate from the same mechanism, so it is unclear whether they
 are separable, even in principle. Further research is needed to fully understand the underlying mechanism of both
 obligatory CO and CO interference.

To include data from chromosomes with 0 or 1 crossover, the authors introduce analysis based on “missing pairs.”
 As the authors note, the definition of d_{mis} is critical because its contribution to Lint directly scales with the number
 of missing pairs (i.e. $1 - \varphi$). However, the choice of $d_{\text{mis}} = L$, while reasonable, is arbitrary. The authors should
 further describe the rationale behind this choice.

We agree that the choice of d_{mis} is important for our metric, and that other definitions of d_{mis} are conceivable.
 However, values larger than L are questionable since such separation distance cannot be observed. In contrast, values
 smaller than L restrict the possible values of L_{int} to smaller numbers, and thus limit its informative value. We thus
 conclude that $d_{\text{mis}} = L$ is the most natural choice. We revised the main text to clarify this rationale.

Related, the authors should report values of ϕ across the analyzed datasets to help evaluate the impact of this
 contribution on reported values and for the improvement in precision.

Since different readers might be interested in different aspects of these plots, we now simply provide the raw data
 in a supplementary spreadsheet.

I am confused by the data in Fig. 2A that shows $\approx 18\%$ of chromosomes bear no crossovers. This is in conflict
 with the obligate crossover assumption. Looking at Durand 2022, it seems like these values may be per-chromatid.
 However, isn't the relevant metric for interference the crossovers formed along each stretch of SC? Please clarify.

The referee is correct in pointing out that these values from [1] are per chromatid and therefore do not conflict with
 the obligatory CO. We mention in the main text, and show in detail in the SI (cf. SI-2B), that L_{int} is invariant to
 random sub-sampling, implying that the measurements are not affected by the transition from cytological data from
 bivalents to genetic data from chromatids, provided that the proportion of non-interfering class II COs is negligible.
 We revised the caption of Fig. 2A to stress that we show genetic data.

I am grateful that the authors include the *msh4* data despite it not clearly fitting their framework of SC vs. crossover
 factor perturbation. However, they should briefly note this and possibly speculate about possible explanations in the
 text.

In the *msh4* mutants, the residual CO are class II CO, which are not subject to interference [6] (we added the
 reference to the main text). It is thus expected to observe a reduction of CO interference in genetic data. Note that
 this is of different nature of *zyp1* or *syp-4*, where we observed a reduction of interference of class I CO.

While the authors correctly note that their results are consistent with the coarsening model, alternate plausible
 models based on the chromatin or SC itself can produce results quantitatively consistent with crossover outcomes
 (e.g. Zhang PLoS Genetics 2014). The existence of alternate models (and ideally, whether/how Lint or other metrics
 could in the future be used to analyze them or discriminate between alternatives) should be briefly noted in the text.

We agree that alternative models are possible. Indeed, we are convinced that L_{int} will allow discriminating between
 models, particularly when time-courses become available in experiments. We revised the discussion to stress this

point.

C. Suggestions for presentation:

Some colors/fit lines in Fig S1 either do not match the caption or are missing.

Thank you for noticing this. We have adapted the caption of Fig. S1 to match the colours used.

It is difficult to distinguish between members of the orange and of the purple families of colors in Fig 6B and S3.

We revised the color scheme used in Fig. 6B and Fig. S3 to make it easier to distinguish between members of different families.

To improve uptake across the community and help the reader get started with the provided code, I would recommend providing a schematic or pseudocode of how L_{int} is calculated from data as part of the supplementary material. However, this is not critical.

We improved the README to help the reader get started with the provided code.

D. Reviewer #1 (Remarks on code availability):

Only briefly reviewed. In general, the raw code seems reasonable and brief enough to be parsed by the reader. As hosted, `measure_CO_interference.html` did not produce results (clicking "calculate quantities" after loading the example data gave no response on Chrome or Firefox).

Thank you for notifying us about the problems of running the code. We were able to reproduce the problem and traced it to a bug in `codeocean.com`. Briefly, we use an HTML form to allow users to input custom data, which is then processed using JavaScript to calculate L_{int} . However, the `codeocean.com` webpage disables forms in webpages, rendering our code non-functioning. A work-around is to download the file `measure_CO_interference.html` and open it locally.

Because of these problems, we decided to opt out of the `codeocean.com` option and instead provide the HTML file `measure_CO_interference.html` as a supplementary file. We will also upload our code to `github.com` and link this in our final version of the manuscript.

II. REVIEWER #2 (REMARKS TO THE AUTHOR):

This article derives two new metrics to measure aspects of crossover interference. While they in some ways provide similar information to L_{CoC} (L_{int}) and the gamma shape parameter ($L_{\text{int}}^{\text{norm}}$) their main advantage lies in the fact that they include data from all COs, account for chromosomes with 1 or no COs, do not require any binning or parametrization and are not affected by sub-sampling. As a result, the derived metrics show improved precision over L_{CoC} and gamma. The determination of the null expectation (d_{noInt}) from observed CO positions also means that non-uniform crossover distributions are taken into consideration.

This work will be of interest to the meiosis community and is very timely, as there is considerable interest currently in modelling crossover interference and testing several competing models. This new metric will likely assist in the interrogation of these models.

There are now quite a few metrics of interference, each of which has their pros and cons. This metric too, has its pros and cons. It has numerous advantages (outlined above) in terms of its calculation and making maximal use of the available data – though these do appear to introduce their own issues e.g. the somewhat arbitrary scaling of L_{int} with L when interference is strong. On balance though, I think it is a very good contribution to the field and will likely gain use over time.

In general, the article is quite well written, though I have suggested several amendments below to improve understanding and readability.

Methodology is largely sound (with some points listed below), though it would be good to provide some more detail on some of the assumptions and strategy for dealing with cytological data (where specific chromosomes are usually not identified) to aid reproducibility (see below).

We thank the referee for their summary of our work and the positive assessment.

A. Specific points:

Abstract: “We show that it faithfully captures known aspects of crossover interference and provides superior
statistical power over previous methods.” As far as I can see only one statistical comparison to previous method is
provided in the main body of the text. If multiple statistical comparisons are referenced in the abstract, then they
should all be mentioned – and ideally covered in some depth – in the main text.

The two methods we refer to in this sentence are the interference distance L_{CoC} based on the coincidence coefficient
CoC and the gamma shape parameter ν . In the main text we show that the interference length L_{int} has several
advantages over these, which are already covered in some detail. We have adapted the text accordingly. We also
added a more comprehensive comparison of the various quantities in Table S1 in the Supporting Material.

Line 39: Consider re-wording. “Bias” is somewhat pejorative here. Cytological data and genetic data provide
different information. Depending on your interest, one or the other set of data (and the interference metrics derived
from them) may be more or less useful. For understanding the mechanistic basis of crossover interference, cytological
data (which lacks data from non-interfering COs) is likely to be more informative. For understanding patterns of
inheritance, then a metric that takes into account non-interfering crossovers is likely to be of more interest.

This is a fair point! We have changed the wording to ”systematic difference”.

Line 101+: You state “ d_{int} quantifies observed distances”, I think this need some further elaboration here, even
if full details are given further down in the text. Presumably this is this some form of normalised average inter CO
distance?

In essence, d_{int} is the average distance of all observed CO pairs, including the missing pairs with an associated
distance of $d_{mis} = L$. The details of this definition are subtle, and we thus expand on these in two separate paragraphs.
To reveal this structure more transparently, we now state this aspect explicitly in the revised text.

Line 128: How then are chromosomes with 0 or 1 CO taken into account? The next paragraph needs to be better
linked so that the reader knows this information is coming.

Chromosomes with 0 or 1 COs are taken into account by their effect on the average CO count $\langle N \rangle$ and thus by
influencing the number of missing pairs as explained in the following paragraph. We realized that the two paragraphs
were not connected very well, and we thus revised this part of our text.

183 / Figure 3B: I do not fully understand how to interpret this figure. As a non-mathematician it is not intuitive
to me why the blue area between the two cumulative distributions corresponds to L_{int} . Some more detail is needed
– I imagine many of the researchers most interested in this publication will also not be mathematicians. . .

Fig. 3B essentially shows the cumulative distribution function, which describes the frequency of values in the
distribution that are smaller than a certain inter-CO distance. The cumulative distribution is obtained by integrating
the probability density that led to the histogram shown in Fig. 3A up to a certain length. Consequently, binning is
not necessary for obtaining the cumulative distribution function and L_{int} corresponds to the blue area between the
curve. We have revised the caption to stress that panel B visualizes the same data as panel A, and we stressed that
the cumulative distribution showed directly that calculating L_{int} does not require binning.

Line 243: SI-2E should this be SI-2F?

The original reference to section SI-2E was correct. The scenario of *complete interference* refers to the one CO
limit, which is described in SI-2E. We have changed the last sentence in SI-2E to improve clarity.

It is unclear to me if L_{int} can incorporate data from different chromosomes where d_{mis} differs or must L_{int}
be calculated independently for different chromosomes. How is d_{mis} dealt with for cytological data where different
chromosomes have different lengths but are not uniquely identifiable? Or when even the same chromosome will usually
have different lengths in different cells.

We indeed compute L_{int} individually for each chromosome. For cytological data, we assigned the data to the
individual chromosomes by ranking their respective SC lengths, if the assignment is not explicit in the respective
data source. For analysis, we normalized the varying lengths of data for the same chromosome in different cells to
the average SC length measured, which provides a sound definition of d_{mis} . In the revised text, we now stress that
we use the average SC length for cytological data, and we included more details of the ranking procedure in the data
handling section in SI-3.

In Figure 4B, I am very surprised that similar values are obtained for L_{int}/L genetic and L_{int}/L cytology, given
that the latter only includes non-interfering (e.g. MLH1 marked) COs, while the former includes both interfering and
non-interfering crossovers. I would have expected L_{int}/L genetic to be systematically lower than L_{int}/L cytology.

Indeed, this seems to be the implication given the tomato data shown in figure 5B. In this case, L_{int}/L would be
 systematically lower when including both class I and class II crossovers. Why then are these estimates equivalent?
 Has some correction been made to account for class II crossovers in this comparison e.g. Supp 2H?

If a correction has been applied, then a) this needs to be made much more explicit, as the results as presented are
 very misleading b) the comparison of uncorrected genetic / cytological estimates needs to be shown instead of (or in
 addition to) the corrected comparison

If a correction has not been applied, then can the authors provide some explanation as to why the two estimates
 are not different and why this appears to contrast with the tomato data shown in figure 5B. Is this something to do
 with subsampling for genetic data (but not for the cytological class II data from tomato)? Or something to do with
 the different backgrounds? E.g. pure inbred lines for cytology vs hybrids for genetic data? I notice in Figure 4A that
 crossover counts from genetic data are not systematically higher than crossover counts for cytological data. . .

No correction was applied here. Indeed, Fig. 4A shows that the CO count in the genetic data does not systematically
 deviate from the prediction that it is half of the foci count in the cytological data. This is consistent with the low
 fraction of class II COs in *A. thaliana*, which is estimated at 15% [7, 8]. Nevertheless, the fact that different genetic
 backgrounds are used here could have some effect and counterbalance a possible low proportion of class II COs in
 Fig. 4A. Given this negligible difference of CO count, we deemed a correction not necessary. To improve clarity of the
 main text, we now mention explicitly that class II COs seem negligible for *A. thaliana* and human. If class II COs
 were more frequent and the distribution was known, one could apply a correction, as proposed in SI-2H. Apparently,
 this is the case for tomato, and we thus analyzed this correction; see main text describing results in Fig. 5B.

 It strikes me that there are many assumptions going on here that require some further explanation/consideration in
 the text. For example, the comparison Fig 4B requires cytological data to be matched to specific chromosomes. My
 understanding is that in the published data the specific chromosomes on which crossovers are formed have not been
 identified. How then is this choice then made? Presumably the longest chromosome measured by μm SC is always
 assumed to the longest chromosome as measured in Mb?

For cytological data we assigned the data to the individual chromosomes by ranking their respective SC lengths
 if the assignment is not explicitly provided in the respective data source. The procedure is described in the data
 handling section SI-3.

 Similarly, L_{int} takes CO positioning into account (when determining d_{point}) which has polarity when analysing
 genetic data based on the reference chromosome sequence. But presumably does not for cytological data where this
 would require specific labelling of one of the chromosome ends. How is this handled? Is the end which represents the
 start position randomly chosen?

We do not take polarity into account when analysing cytological data and use the start position as given in the data
 sources. We agree with the referee that this could be a potential issue for strongly asymmetric data, and mention this
 issue in the revised data handling section SI-3.

 What is d_{mis} for cytological data? Does it take into account the fact that there is variation in total SC length
 between cells? E.g. by averaging the lengths of only chromosomes with 0 or 1 CO to give d_{mis} ?

Our proposed procedure is to calculate L_{int} individually for each chromosome. For the analysis, we normalise the
 different lengths of the data for the same chromosome in different cells to the average SC length measured, providing
 a sound definition of d_{mis} . We now stress in the revised text that we use the average SC length for each chromosome
 when analyzing cytological data and thus neglecting the possible influence of SC length variation (c.f. [9]) on the CO
 interference quantified by L_{int} .

 Line 372-378: One question I had when reading this section was “Is this difference associated with proportion of
 total crossovers contributed by non-interfering class II COs?” I then realised this is exclusively cytological data and
 so is class I only. The authors may like to flag up this point somewhere in this paragraph.

We revised the initial sentence of the paragraph to stress that cytological data only contains class I COs.

 Fig 5: The fact that L_{int} scales with L is somewhat of a drawback to the metric as I assume (and other modelling
 suggests) that mechanistically, interference is implemented over the same distance of SC on all chromosomes within
 a cell. Is this scaling primarily due to the fact that $d_{\text{mis}} = L$?

The interference length does indeed scale with L for strong interference scenarios. However, species with L_{int} much
 lower than $\frac{2}{3}L$ show only a weak dependence on chromosome length L ; see Fig. 5 and the accompanying discussion
 in the main text. In essence, L_{int} is weakly dependent of L if there are many interfering COs, in accordance with the
 picture of the referee. In contrast, for strong interference, L_{int} is limited to L since both d_{obs} and d_{mis} scale with L .
 The fact that L_{int} is limited to L essentially follows from the impossibility of observing CO separations larger than

L , so it is in principle not possible to observe L_{int} larger than L .

Minor point – what are used as the proxies of SC length in mutants (such as *zyp1*) which lack an SC? Is this just
axis length?

For the *A. thaliana zyp1* mutant we assign the SC length of the respective wild-type data. This is consistent with
the observation that the axis length is not changed significantly in the *zyp1* mutant [1]. We clarified this in the revised
data handling section SI-3.

Figure 6A: I find there is far too much going on in this Figure. I would prefer four separate panels with more of
a zoom / of the box in the top left corner (for both A and B), so that the data can actually be seen. In the text a
point is made about relative fits of simulations for low N . It is very hard to assess this with the given plots.

We agree that this is a dense plot, but we think that it is helpful to show all this data in one plot to emphasize
similarities. However, we acknowledge that not all details are visible. Since different readers might be interested in
different aspects of these plots, we now simply provide the raw data in a supplementary spreadsheet.

Fig 6B: the colours are very similar for *zyp1* and *msh4*. Alternative colours should be used so that these can be
distinguished.

We revised the color scheme in Fig. 6B and Fig. S3 to make it easier to distinguish between members of different
families.

It is interesting here that *C. elegans* has a low(ish) value of $L_{\text{norm.int}}$ given complete interference in this species.
Presumably for $N \sim 1$, then $\varphi = 0$ so $L_{\text{int}} = L - d_{\text{point}}$. Why is there such variation in $L_{\text{norm.int}}$ for cases where
$N \sim 1$? Is this entirely due to differences in CO distribution causing different values of d_{point} ? This seems to me a
limitation of the metric. I.e. quite different values of $L_{\text{norm.int}}$ for different cases of strong interference.

For *C. elegans* wild-type data the low value can indeed be explained solely by the differences in CO distribution.
In case of the *C. elegans syp-4* mutant the value of L_{int} is different also because of different average CO count $\langle N \rangle$.
We now mention this point in the revised main text.

I imagine (though am happy to be corrected!) that a draw back of the L_{int} metric is that the same value could
be derived from situations where only a few COs occur but there is no interference, and situations where many more
crossovers occur but interference is present.

We agree that systems with different $\langle N \rangle$ can exhibit similar L_{int} . This is a features since we designed L_{int} and
$L_{\text{int}}^{\text{norm}}$ with the aim to capture aspects of CO interference that are orthogonal to CO count $\langle N \rangle$. To interpret a given
situation, it is thus advisable to analyze $\langle N \rangle$ and L_{int} .

Where would *msh4* data (few COs, no interference) be, were it shown on figure 5?

We agree that showing additional data is useful to investigate CO interference in more detail. We decided to show
additional plots in the style of Fig. 5 in the Supporting Information as Fig. S4. The *msh4* data does not seem to
deviate significantly from wild type.

Line 461: Is this both for cytological and genetic data?

Yes. As highlighted in the previous paragraph, $L_{\text{int}}^{\text{norm}}$ does need to account for the factor of 2 for random sub-
sampling between cytological and genetic data.

My take here is that $L_{\text{norm.int}}$ is akin to the gamma model where it tells us about the regularity of inter CO
distance, irrespective of absolute inter-crossover distance? I would find a comparative table useful which lays out the
respective attributes of gamma, L_{coc} , L_{int} , $L_{\text{norm.int}}$ and their interpretation.

We agree with the referee that a more comprehensive comparison of the various quantities could be helpful and we
thus added on as Table S1 in the Supporting Material.

Line 500: “cluster” is quite generous here... the values of $L_{\text{norm.int}}$ appear highly variable in these mutant
contexts.

We agree that “cluster” is not the right term and adjusted the language in the revised manuscript.

Other points:

Discussion of the contribution to / effect of class II crossovers on L_{int} and $L_{\text{norm.int}}$ seems limited. It would be
good to see how L_{int} and $L_{\text{norm.int}}$ respond in situations where there are high levels of class II COs (e.g. *recq4*
mutants in *Arabidopsis*). Similarly, I think the manuscript would benefit from running simulations to explore how
L_{int} / $L_{\text{norm.int}}$ respond to differing proportions of class II crossovers e.g. coarsening simulations with different
added proportions of randomly distributed COs.

We agree that detailed analysis of mutants is interesting. Indeed, we envision our measure to be used for such
 analysis by the community in the future. Concerning the influence of class II COs, we refer to the section SI-2H for
 a more detailed discussion. Briefly, a higher fraction of class II COs reduces the values of L_{int} and $L_{\text{int}}^{\text{norm}}$. The effect
 of a certain ratio of class II COs can be described by the simple equation

$$L_{\text{int}} = \varphi^2 L_{\text{int}}^{\text{I}},$$

where $\varphi = \langle N_{\text{I}} \rangle / \langle N \rangle$ denotes the fraction of class I COs and $L_{\text{int}}^{\text{I}}$ is the pure interference length measured for only
 class I COs.

 A somewhat leftfield thought, but I wondered if L.int is able to give any meaningful output from F2 data? It would
 clearly underestimate L.int and be sex-averaged, but perhaps it might respond in a predictable manner (e.g. half the
 sex average - this could be tested with simulations / theory) and might possibly enable some insights into interference
 to be gained when backcross data are not available. If so, this would give added strength to the utility of this new
 metric.

This is an interesting suggestion for further research where L_{int} might be useful.

 There is some inconsistency in which data are shown for which analyses. E.g. for some both cytological and genetic
 are shown, for others just cytological (e.g. Fig 5). Some mutants are shown in some figures/analyses but not in
 others. There are some legitimate reasons for this, even if simply to prevent charts from becoming overcrowded. But
 to avoid any sense of “cherry-picking” which data are shown, it might be better to have (even if just added to Supp
 Material) all analyses/charts shown for all data used in the manuscript.

We added Fig. S4 in the supplementary material showing the genetic data comparable to Fig. 5. All other data is
 already shown wherever possible.

 As mentioned above, I think it would be useful to have a comparative table that lays out the respective attributes
 of gamma, L.coc, L.int, L.norm.int and their interpretation, pros/cons e.g. affected by sub-sampling, etc, etc

We agree with the referee that such a comparison table would be useful and we thus added it as Table S1 to the
 Supplementary Material.

B. Reviewer #2 (Remarks on code availability):

I have tried using the code supplied via Code Ocean. I was able to get the html page working and this seemed to
 give good results. I could not get the python script to work - perhaps a README file would be useful here. A quick
 scan suggests it is written specifically for Arabidopsis thaliana data. If so, a general use version may be more helpful.

The python functions are written for generic data. However, we for convenience supply exemplary data of *A.*
 *thaliana* that can be analyzed directly. We revised the code to separate the general functions calculating L_{int} from
 the script which analyzes the exemplary data. We also improved the README file.

III. REVIEWER #3 (REMARKS TO THE AUTHOR):

During meiosis, crossovers promote reciprocal DNA exchange between paired homologous chromosomes. Conse-
 quently, the magnitude of DNA exchange is determined by the position and number of chromosomal crossovers,
 which are directly influenced by the conserved phenomena of crossover interference (the inhibition of closely-spaced
 crossovers). Despite crossover interference being studied for over a century, the meiosis field has yet to reach a con-
 sensus on a standard approach for reliably quantifying crossover interference. Here, Ernst, Mercier and Zwicker point
 out inadequacies in current methods for quantifying interference (CoC and gamma shape parameter analysis) and
 formulate a new metric (Interference Length) that more comprehensively captures a wide variety of meiotic crossover
 patterning features in a single number.

The authors describe in detail how the Interference Length metric has been derived and clearly justify the ratio-
 nale for including particular values. They then demonstrate and explore the functionality of Interference Length by
 comparing values obtained using publicly available cytological and genetic datasets in different organisms and genetic
 backgrounds. They conclude that Interference Length offers a useful method (superior to CoC or gamma) for quanti-
 tatively comparing aspects of crossover interference both within and across species. Finally, they discuss how inferences
 drawn from these Interference Length comparisons are consistent with the recently proposed coarsening model for
 meiotic crossover patterning.

I am convinced that Interference Length offers a superior method for quantifying crossover interference in a single unit compared with previous methods and would be a useful addition to the meiosis field. However, I have a few comments that it would be useful for the authors to address:

We thank the referee for their summary of our work and their positive assessment.

A. Major

1. The authors demonstrate that Interference Length can be used to compare crossover patterning data between organisms and how this could be used, for example, to support a common underlying mechanism (e.g. the coarsening model). However, the calculation required to produce an Interference Length value is complex and it is not always intuitive how this value should be directly interpreted with regard to the crossover patterning factors that directly underlie changes in the value (i.e. why are normalised interference length values in *S. cerevisiae* lower than in other species?). Therefore, the case of *S. cerevisiae* would benefit from further discussion and analysis of why this species is an exception (presumably this is a consequence of the chromosomes being very small and having a high crossover frequency?).

We agree that our defined L_{int} is not always intuitive. In essence, this is an unavoidable side effect of constructing a scalar measure of a complex phenomenon. However, we maintain that L_{int} provides more robust information than alternative measures, like d_{CoC} or ν of the Gamma distribution. To clarify this better, we added a comparison table of all these measures as Table S1 in the revised supplement. In particular, scalar measures, like L_{int} , enable the comparison of many datasets to identify interesting aspects, like the lower values of L_{int} for *S. cerevisiae*, consistent with literature which also suggest weak CO interference [10, 11]. The detailed origins of such deviations may then only be identified by a more detailed analysis, taking into account more information than what L_{int} provides.

I believe cytological Zip3 foci data is available for three *S. cerevisiae* chromosomes in Ref 43. Perhaps including this data in Fig 5 and 6 would also help to clarify the cause of this difference.

We agree that analyzing the data provided in Ref. 43 could shed light on the behavior of *S. cerevisiae*. However, we do not believe that an extended discussions of CO interference in yeast is helpful in our paper, which focuses on a novel measure of CO interference and its principle behavior. We instead envision that a future publication, dedicated to analysis of *S. cerevisiae* data could illuminate CO interference in that organism using L_{int} , $\langle N \rangle$, and more detailed measures.

2. I'm still not completely convinced by the authors' claims that "cytological and genetic data can be compared directly" using Interference Length. I understand that Interference Length is invariant to subsampling, but I think there are other points to consider to enable direct comparison.

There are indeed further assumptions necessary for direct comparison of cytological and genetic data. First, the comparison assumes a linear and uniform compaction of the DNA along the SC. Second, it assumes that the ratio of class II COs is either negligible, or – if we apply the correction given in SI-2H – the relative distribution of class II COs along the axis is the same as for class I COs. Both assumptions are simplifications which might affect the direct comparison. We revised the main text to make these assumptions more explicit.

Surely the presence of heterochiasmy (and the authors analysis of this) argues against the suggestion that cytological data and genetic data can be directly compared? The authors plot normalised interference length from genetic and cytological data in Fig 4B and demonstrate agreement for human male interference length. However, if they were able to add human female data to the plot in Fig 4B I doubt this would agree as there would be a change in the normalisation factor for SC length but not for chromosome length. Similarly, plot Fig 4C and Fig 4D demonstrate that measuring Interference Length using genetic data or cytological data can result in very different interpretations (Interference Length is smaller when measured genetically for human males compared with females, but equivalent when measured cytologically).

We agree that the compaction factor, which converts SC lengths into genetic lengths, is different between male and female, and eventually leads to heterochiasmy. However, this difference has no bearing on L_{int}/L , since both the numerator and denominator are scaled by the same factor. There might be secondary effects leading to deviations in L_{int}/L , e.g., non-uniform compaction or significant class-II COs, but we have no evidence that these are relevant.

Am I also correct in thinking that genetic Interference Length values in Fig 4 have not been corrected for the presence of class II COs (which as the authors' mention, is a hurdle for interpreting interference in genetic CO data)?

The referee is correct in that the genetic interference length in Fig. 4 is not corrected for the possible presence of class II COs. In particular, Fig. 4A shows that the CO count in the genetic data does not systematically deviate

from the prediction that it is half of the foci count in the cytological data. This is consistent with the low fraction of
 class II COs in *A. thaliana*, which is estimated at 15% [7, 8]. Given this negligible difference of CO count, we deemed
 a correction not necessary. To improve clarity of the main text, we now mention explicitly that class II COs seem
 negligible for *A. thaliana* and human. If class II COs were more frequent and the distribution was known, one could
 apply a correction, as proposed in SI-2H.

 On a related note, is the genetic data in Fig 6 scaled with respective SC lengths (and thus measured in SC space)
 to enable comparison with cytological data? I would have thought the scaling of genetic data (or not) would influence
 the impact of heterochiasmy in comparing male and female genetic data in this plot.

The data in Fig. 6 is not scaled with the respective SC lengths since the definition of $L_{\text{int}}^{\text{norm}} = \frac{L_{\text{int}}}{\langle N \rangle L}$ does normalise
 the measured L_{int} with the respective chromosome length (either in μm or Mb-space).

 A final point to note is that the scaling of genetic data by SC lengths assumes that compaction of chromatin along
 meiotic pachytene chromosomes is uniform. However, this is not the case as compaction (i.e. μm of SC per Mb of
 DNA) varies significantly between e.g. chromosome arms and pericentromeric regions.

We agree that our comparison between genetic and cytological data relies on uniform compaction. We now stress
 this explicitly in the revised text. If the non-uniform compaction is known, it can be applied to the CO positions
 directly, and a refined L_{int} can be determined. However, we are not aware of detailed measurements of the compaction
 in conjunction with CO analysis.

 3. It is surprising that the normalised Interference Length values for arabidopsis deviate so much from the predicted
 values from the coarsening model (Fig 6), given that the coarsening model was parameterised using arabidopsis data.

The coarsening model in [1] was fitted for quantitative agreement in CO count. The remaining quantifications
 (CoC curves and histograms of inter-CO distances) in [1] then showed satisfactory agreement. The model was not
 optimized to fit L_{int} , so that a deviation is not very surprising. Note also that [1] assumed a uniform distribution of
 initial foci along the SC, in contrast to [3], which might affect L_{int} . We think it will be interesting to investigate in
 the future what aspects of the model affect L_{int} significantly.

 I think further exploration of this deviation is warranted (especially as the authors conclude that their data is
 consistent with the coarsening model). What parameters within the model would the authors alter to achieve a better
 fit of the simulated Interference Length to the experimental arabidopsis data? Do simulation outputs from other
 versions of the model (e.g. from Ref 9) result in better fits to the data?

We agree that a more detailed analysis of the coarsening model is interesting and should be done. However, it is
 beyond the scope of the present manuscript, which focuses on introducing L_{int} as a new quantity, and analyzes its
 basic properties. In principle, there are many parameters and aspects of the coarsening model that could be adapted:
 non-uniform initial foci density or SC density, non-uniform transport properties of the SC, exchange with nucleoplasm
 (compare also [2]), time-dependent exchange rates, and other aspects. We are convinced that a detailed analysis and
 extension of the coarsening model deserves a separate publication. We revised the last paragraph of the discussion
 to mention alternative models, how L_{int} could help discriminate between them, and how L_{int} could help in model
 refinement.

 Furthermore, the authors use the parameterisation from arabidopsis to compare to data from many other organisms
 (Fig 6). There is no reason to think that the Arabidopsis parameters will be universal. Hence, for this comparison to
 be rigorous, the authors really need to make some attempt to refit the coarsening model for these other cases. This
 might lead to better agreement between the model and the data.

We agree with the referee that there is no reason to expect that the parameters of Arabidopsis explain CO inter-
 ference in other organisms quantitatively. The focus of the present publication is not on the coarsening model, and
 we simply added it to supply a qualitative comparison to this exciting model.

 4. In the authors' analysis, they set the distance associated with missing pairs $d_{\text{mis}} = L$, the chromosome length.
 While this scaling is reasonable, setting the proportionality constant strictly to one is arbitrary. I wonder whether
 better results could be obtained using $d_{\text{mis}} = \alpha L$, with alpha a free parameter.

We agree that other definitions of d_{mis} are conceivable, but we maintain that $d_{\text{mis}} = L$ is a natural choice. Intro-
 ducing $d_{\text{mis}} = \alpha L$ requires specifying the value of α , and it is unclear how to choose it. Values larger than L ($\alpha > 1$)
 are questionable since such separation distance cannot be observed. In contrast, values smaller than L restrict the
 possible values of L_{int} to smaller numbers, and thus limit its informative value. We have thus chosen $\alpha = 1$, implying
 $d_{\text{mis}} = L$, because we are convinced it is the best possible choice. We revised the main text to clarify this rationale.

B. Minor

5. The authors discuss the benefits of Interference Length compared with CoC and gamma analysis, which both
provide a single convenient metric for quantifying interference. However, it is also possible to plot and measure
individual aspects of CO patterning (CO number, CO spacing, CO position) separately and then compare these
(very easy to understand and interpret) plots individually without the need to compress the information into a single
number. I think the manuscript would also benefit from a discussion of why Inteference Length offers a superior (or
useful complementary) approach compared to analysing individual crossover patterning features in isolation.

We agree that there are different measurements that reveal information about CO interference. The advantage of
a scalar measure, like L_{int} , is that it facilitates direct comparison of multiple species, sexes, and genotypes, which
we stress in the first paragraph of the discussion. Such a broad comparison might reveal interesting aspects of CO
interference, which can then be analyzed in detail using additional measures.

6. Line 546 -547. The authors discuss how species with many COs might abort coarsening early. However, there
are other explanations, such as changes in the dosage of HEI10 (orthologs), or altered binding/unbinding kinetics. I
think these should also be discussed/mentioned.

We agree with the referee, and noted that our sentence was misleading. We revised the main text to clarify that
we are discussing a scenario where coarsening is not yet completed, which could be caused earlier abortion, increased
HEI10 levels, or other changes.

IV. REVIEWER #4 (REMARKS TO THE AUTHOR):

I co-reviewed this manuscript with one of the reviewers who provided the listed reports. This is part of the Nature
Communications initiative to facilitate training in peer review and to provide appropriate recognition for Early Career
Researchers who co-review manuscripts.

We thank the referee for their work.

-
- [1] S. Durand, Q. Lian, J. Jing, M. Ernst, M. Grelon, D. Zwicker, and R. Mercier, Joint control of meiotic crossover patterning
by the synaptonemal complex and *hei10* dosage, *Nature Communications* **13**, 5999 (2022).
- [2] J. A. Fozard, C. Morgan, and M. Howard, The synaptonemal complex controls cis- versus trans-interference in coarsening-
based meiotic crossover patterning, *bioRxiv*, 2022.04.11.487855 (2022).
- [3] C. Morgan, J. A. Fozard, M. Hartley, I. R. Henderson, K. Bomblies, and M. Howard, Diffusion-mediated *hei10* coarsening
can explain meiotic crossover positioning in *arabidopsis*, *Nature Communications* **12**, 4674 (2021).
- [4] L. Zhang, Z. Liang, J. Hutchinson, and N. Kleckner, Crossover patterning by the beam-film model: analysis and implica-
tions, *PLoS genetics* **10**, e1004042 (2014).
- [5] N. Kleckner, D. Zickler, G. H. Jones, J. Dekker, R. Padmore, J. Henle, and J. Hutchinson, A mechanical basis for
chromosome function, *Proceedings of the National Academy of Sciences* **101**, 12592 (2004).
- [6] N. M. Hollingsworth and S. J. Brill, The *mus81* solution to resolution: generating meiotic crossovers without holliday
junctions, *Genes & Development* **18**, 117 (2004), <http://genesdev.cshlp.org/content/18/2/117.full.pdf+html>.
- [7] J. D. Higgins, S. J. Armstrong, F. C. H. Franklin, and G. H. Jones, The *arabidopsis* *mut*s homolog *atmsh4* functions at
an early step in recombination: evidence for two classes of recombination in *arabidopsis*, *Genes & development* **18**, 2557
(2004).
- [8] R. Mercier, C. Mézard, E. Jenczewski, N. Macaisne, and M. Grelon, The molecular biology of meiosis in plants, *Annual*
*review of plant biology* **66**, 297 (2015).
- [9] S. Wang, C. Veller, F. Sun, A. Ruiz-Herrera, Y. Shang, H. Liu, D. Zickler, Z. Chen, N. Kleckner, and L. Zhang, Per-nucleus
crossover covariation and implications for evolution, *Cell* **177**, 326 (2019).
- [10] L. Zhang, S. Wang, S. Yin, S. Hong, K. P. Kim, and N. Kleckner, Topoisomerase ii mediates meiotic crossover interference,
*Nature* **511**, 551 (2014).
- [11] C. Girard, The regulation of meiotic crossovers distribution: a coarse solution to a century-old mystery?, *None* (2023).

REVIEWERS' COMMENTS

Reviewer #1 (Remarks to the Author):

I am satisfied with the authors' reply and the revised manuscript. I support the publication of this work and commend the authors on their valuable contribution to the field.

Reviewer #2 (Remarks to the Author):

The authors have addressed my main concerns. There is a problem with the y-axis label for Figure S4 which does not appear in my version of the PDF. In addition, I still think figure S4B would benefit from adding data from a mutant with increased class II crossovers (e.g. *recq4*). This is an important context in which measures of interference show different outcomes. An analysis of existing data for this context (for which there are many existing data sets), rather than the purely theoretical treatment described in S2H would be of benefit.

Reviewer #3 (Remarks to the Author):

The authors have now comprehensively revised their manuscript and we are now happy to see the paper published in Nature Comms.

Reviewer #4 (Remarks to the Author):

Rebuttal Letter: Interference Length reveals regularity of crossover placement across species

Marcel Ernst, Raphael Mercier, and David Zwicker
(Dated: August 12, 2024)

We thank the editor and the four referees for handling our manuscript and giving thoughtful comments. Below we provide a point-by-point response (blue text) to all the referees' comments (black text). We also mention how we revised the manuscript and provide a PDF highlighting the differences to the previously submitted version.

I. REVIEWER #1 (REMARKS TO THE AUTHOR)

I am satisfied with the authors' reply and the revised manuscript. I support the publication of this work and commend the authors on their valuable contribution to the field.

We thank the referee for their work and their positive assessment.

II. REVIEWER #2 (REMARKS TO THE AUTHOR):

The authors have addressed my main concerns.

We thank the referee for their work and their positive assessment.

There is a problem with the y-axis label for Figure S4 which does not appear in my version of the PDF.

Thanks for pointing this out. We fixed the y-axis label in the revised text.

In addition, I still think figure S4B would benefit from adding data from a mutant with increased class II crossovers (e.g. *recq4*). This is an important context in which measures of interference show different outcomes. An analysis of existing data for this context (for which there are many existing data sets), rather than the purely theoretical treatment described in S2H would be of benefit.

We agree with the referee that this would be interesting indeed. Unfortunately, we are not aware of *recq4* data of sufficient quality that is not referring to F2 populations.

III. REVIEWER #3 (REMARKS TO THE AUTHOR):

The authors have now comprehensively revised their manuscript and we are now happy to see the paper published in Nature Comms.

We thank the referee for their work and their positive assessment.

IV. REVIEWER #4 (REMARKS TO THE AUTHOR):

We thank the referee for their work and their positive assessment.